# RS-Reg: Probabilistic and Robust Certified Regression through Randomized Smoothing

**Aref Miri Rekavandi**                                            *aref.mirirekavandi@unimelb.edu.au*
*School of Computing and Information Systems*
*The University of Melbourne*

**Olga Ohrimenko**                                                 *oohrimenko@unimelb.edu.au*
*School of Computing and Information Systems*
*The University of Melbourne*

**Benjamin I.P. Rubinstein**                                       *benjamin.rubinstein@unimelb.edu.au*
*School of Computing and Information Systems*
*The University of Melbourne*

**Reviewed on OpenReview:** *https://openreview.net/forum?id=AcLlg4J52H&referrer*

## Abstract

Randomized smoothing has shown promising certified robustness against adversaries in classification tasks. Despite such success with only zeroth-order access to base models, randomized smoothing has not been extended to a general form of regression. By defining robustness in regression tasks flexibly through probabilities, we demonstrate how to establish upper bounds on input data point perturbation (using the $\ell_2$ norm) for a user-specified probability of observing valid outputs. Furthermore, we showcase the asymptotic property of a basic averaging function in scenarios where the regression model operates without any constraint. We then derive a certified upper bound of the input perturbations when dealing with a family of regression models where the outputs are bounded. Our simulations verify the validity of the theoretical results and reveal the advantages and limitations of simple smoothing functions, i.e., averaging, in regression tasks. The code is publicly available at https://github.com/arekavandi/Certified_Robust_Regression.

## 1 Introduction

The ongoing competition between attackers and defenders has a long history in cybersecurity. Whenever attackers have gained an advantage, defenders have countered with innovative empirical techniques, fostering a cycle of continuous evolution. In recent times, research into defenses has embraced the concept of certified robustness, seeking assurances that attackers will find it challenging to discover adversarial examples in the vicinity of test samples for deceiving AI systems. While these guarantees are constrained to a certain type of threat model (which may not align with the attacker's actual strategy), they offer precise and reliable bounds crucial for cybersecurity systems. Randomized Smoothing (RS) has emerged as a prevalent strategy in certifying classification models, demonstrating remarkable scalability to large-scale models. In this approach, majority voting is performed on samples drawn around a given test query. With the greater number of samples drawn, as the estimated parameters tend to their true values, the likelihood of discovering any adversarial examples in the vicinity of the test query to deceive the model goes to zero. RS has been recently extended into other settings such as classifiers with discrete or variable-size inputs (Huang et al., 2023), classifiers with semantic certificates (Fischer et al., 2020), and probabilistic certificates (Pautov et al., 2022b). However, to the best of our knowledge, the framework has not yet been fully expanded into conventional regression problems, while many attacks have been demonstrated (Chiang et al., 2020; Liu et al., 2022). To extend RS to regression, the adversarial robustness definition requires some adjustment. In this study, we

introduce a probabilistic definition for robustness in regression models and subsequently derive performance guarantees, given mild conditions. This new framework allows defenders to pursue probabilistic certificates and strikes a balance between the precision and the amount of effort an attacker must exert to discover an adversarial example. We summarize the main technical contributions of the paper as follows:

- We introduce a variant of probabilistic certification for regression problems where the output variable is multivariate and continuous.

- Using the new definition, we then derive a probabilistic certified upper bound on the input perturbation for a base regression model.

- We then demonstrate that, asymptotically, the output of RS with averaging follows a normal distribution, allowing the probability of obtaining valid results from a smoothed regressor to be determined through an integral over a neighborhood of such a normal density.

- Under mild conditions, we propose a lower bound on the probability of observing valid outputs using the smoothed regressor. Then we find the upper bound on the adversarial perturbation in the input space which satisfies the user-defined probability of observing valid outputs utilizing the smoothed regressor.

The technical progression of the paper is as follows: (**i**) We first examine the robustness of the base regression model, i.e., $\mathbf{f}_\theta(\mathbf{x})$ and derive an upper bound on the $\ell_2$ norm of the adversarial perturbation which satisfies our definition of robust regression. (**ii**) Using an average function as the smoothing policy, we study the asymptotic behavior of the output as $n \to \infty$ (where $n$ is the number of samples drawn from a Gaussian used to perturb the input and then averaged to compute the output) and show that is normally distributed. (**iii**) We then use the results of the base regressor and derive a lower bound probability for regression models with bounded outputs. The results up to this point are valid only in the asymptotic regime, hence, we derive a different bound for the case where the output is bounded and the user considers a discount on the validity range of output variables. This new result is valid for finite-sample scenarios.

## 2 Related Works

RS has seen little application outside classification. Among the few works that extend the RS framework, Pautov et al. (2022a) proposed smoothed embeddings that extend RS to few-shot learning models that map inputs to normalized embeddings. The work considered the case where the $\ell_2$ norm of the embedding is set to one and showed that the smoothing function is Lipschitz in the $\ell_2$ norm. This setup is much different than ours because we do not consider any constraint on the output except a bounded range, and outputs can be independent of each other with different scales. Furthermore, our framework is probabilistic. Salman et al. (2019) addressed the certification for soft classifiers, where the output variables are continuous. However, they were scaled such that they satisfy $\sum p_i = 1$ to finally classify inputs, not to regress any continuous variable. Their results are not probabilistic in the sense that we present in this work. A recent study by Hammoudeh & Lowd (2023) in the context of poisoning attacks and certified ML has been performed for the first time in regression tasks to find a guarantee on the number of training instances that can be inserted into or deleted from a training set without violating the output constraints. The most related work to our study is by Chiang et al. (2020) where the object detection problem has been treated as a regression problem. The result relies on expanding the range of classification models which limits certification to regression models with Softmax activation function in the last layers.

## 3 Preliminaries

**Notation.** In this paper, the base regression model parametrized with $\boldsymbol{\theta}$ is defined such that $\mathbf{y} = \mathbf{f}_\theta(\mathbf{x})$ : $\mathbb{R}^d \to \mathbb{R}^t$, where $d$ and $t$ are the input and output dimensions, respectively. $\mathcal{N}(\mathbf{m}, \sigma^2\mathbf{I})$ indicates a multivariate normal density with mean $\mathbf{m}$ and covariance $\sigma^2\mathbf{I}$, where $\mathbf{I}$ is the identity matrix. The $\ell_2$ norm of a vector $\mathbf{x}$ is denoted by $\|\mathbf{x}\|_2$ and it is defined by $\|\mathbf{x}\|_2 = (\sum_i \mathbf{x}_i^2)^{1/2}$, where $\mathbf{x}_i$ is the $i^{th}$ element of the vector $\mathbf{x}$. The

neighborhood centered around variable $\mathbf{z} \in \mathbb{R}^s$ with radius $\epsilon$ with respect to a given dissimilarity function is denoted by $\mathbf{N}(\mathbf{z}, \epsilon)$. This neighboring function defines a region such that

$$\mathbf{N}(\mathbf{z}, \epsilon) = \{\mathbf{z}' \in \mathbb{R}^s \mid \text{diss}(\mathbf{z}, \mathbf{z}') \leq \epsilon\}, \tag{1}$$

where $\text{diss}(\cdot, \cdot)$ in general, can be any metric or function that the user is interested in and in particular, $\text{diss}_x$ (or $\text{diss}_y$) indicates dissimilarity in the input (or output) space (Miri Rekavandi et al., 2024). $\mathbb{P}\{\text{event}\}$ indicates the probability that an event occurs and $\mathbb{E}\{\mathbf{z}\}$ indicates the expected value of the random variable $\mathbf{z}$. $[t]$ indicates the set $\{1, 2, \cdots, t\}$ and $\Phi(\cdot)$ is the standard Gaussian CDF. $\lceil \cdot \rceil$ rounds up to the nearest integer value. Indicator $\mathbf{1}_{\text{condition}}$ returns 1 only when the condition is *True* and otherwise is zero.

**Randomized Smoothing.** RS is among the certification techniques that can be used for large-scale (and arbitrary) models, given only black-box access to model evaluations, and has been used for the first time in seminal works Cohen et al. (2019); Lecuyer et al. (2019) for classification tasks. In particular, Cohen et al. (2019) showed that if the initial classifier $\mathbf{f}_\theta(\mathbf{x})$ is robust under Gaussian noise, then the new classifier $g(\mathbf{x}) = \arg\max_{c \in \mathbf{Y}} \mathbb{P}(f(\mathbf{x} + \mathbf{e}) = c)$, $\mathbf{e} \sim \mathcal{N}(\mathbf{0}, \sigma^2 \mathbf{I})$ is certifiably robust against an $\ell_2$ norm adversary with radius $\epsilon = \frac{\sigma}{2}\left(\Phi^{-1}(\underline{p_A}) - \Phi^{-1}(\overline{p_B})\right)$ where $\underline{p_A}$ and $\overline{p_B}$ (s.t. $\underline{p_A} \geq \overline{p_B}$) are the lower bound probability of major class, and upper bound probability of runner-up class, respectively. Later studies (Yang et al., 2020; Kumar et al., 2020) showed that for $\ell_p$ norm attacks ($p > 2$), these radii decrease as $\mathcal{O}(1/d^{\frac{1}{2} - \frac{1}{p}})$ suggesting that for other norms and high-dimensional data points, these certificate radii tend to zero showing lack of meaningful insights. RS has mainly relied on Gaussian smoothing, however, improvements have been observed using other smoothing functions such as with Uniform (Lee et al., 2019), Laplacian (Teng et al., 2019), and non-Gaussian (Zhang et al., 2020) smoothing to deal with general types of attacks. We refer readers to Li et al. (2023) for more details on RS and certified robustness.

**Threat Model.** We assume attackers with full information about the regression model and its underlying process, and who are limited to perturbations of an input sample $\mathbf{x}$ within an $\ell_2$-bounded norm. The attacker's ultimate goal is to generate sample $\mathbf{x}' \in \mathbf{N}_x(\mathbf{x}, \epsilon_x)$ that generate an output with large deviation from the $\mathbf{f}_\theta(\mathbf{x})$ where this deviation is likely beyond the user's tolerance threshold. Although this paper focuses on the $\ell_2$ threat model, one can extend the results to other norms such as $\ell_1$, $\ell_p$, and $\ell_\infty$, either in the input or feature space. An extension of the results presented in this paper for the $\ell_p$ attack in the input space can be found in Miri Rekavandi et al. (2024), where a similar problem is explored. Additionally, one could explore scenarios where perturbations are applied to the parameter space in convolutional kernels, such as in blurring or sharpening operators in the context of image modality (Brückner & Lomuscio, 2024).

## 4 Probabilistic Robustness Certification for Regression

Randomized smoothing and corresponding certificates for classifiers are tied to aggregated predictions by majority vote (Mohapatra et al., 2020), which does not apply to regression. We therefore start by defining a new notion of probabilistic certificate that is suitable for regression models.

**Definition 1:** (Probabilistic Robustness Certificate). *Given an example $(\boldsymbol{x}, \boldsymbol{y})$, a (possibly) randomized regression function $\boldsymbol{g}(\boldsymbol{x}) : \mathbb{R}^d \to \mathbb{R}^t$ is said to be certifiably robust with probability $0 \leq P \leq 1$ in the randomness of $\boldsymbol{g}$, with respect to the given input and output dissimilarity functions with radii $\epsilon_x$, $\epsilon_{y_1}, \cdots, \epsilon_{y_t}$, if $\forall \boldsymbol{x}' \in \mathbf{N}_x(\boldsymbol{x}, \epsilon_x)$*

$$\min_{i \in [t]} \mathbb{P}\{\text{diss}_y(\mathbf{g}(\mathbf{x}')_i, \mathbf{y}_i) \leq \epsilon_{y_i}\} \geq P. \tag{2}$$

**Remark 1**: The definition of certified robustness for the function $\mathbf{g}(\mathbf{x})$ differs from the definition of *local Lipschitz function* as $\forall \mathbf{x}' \in \mathbf{N}(\mathbf{x}, \epsilon)$, $\|\mathbf{g}(\mathbf{x}) - \mathbf{g}(\mathbf{x}')\| \leq L(\mathbf{x})\|\mathbf{x} - \mathbf{x}'\|$ familiar from analysis. Our definition is probabilistic: it is acceptable to fall outside the predefined neighborhood, and our dissimilarity function is considered to be general not limited to a certain norm. On the other hand, while the neighborhood region in the input space is defined for all the dimensions simultaneously, in this definition output dimensions are analyzed separately using the neighboring function $\mathbf{N}_\mathbf{y}(\mathbf{y}, \epsilon_y) = \prod_{i=1}^{t} \mathbf{N}_y(\mathbf{y}_i, \epsilon_{y_i})$.

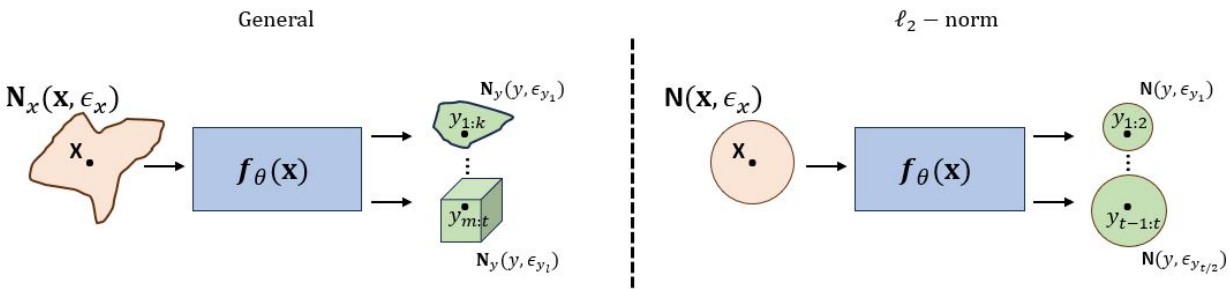

Figure 1: The general schematic of probabilistic certified robustness in regression where input can deviate from **x** in any direction (bounded with respect to diss$_x$ used in $\mathbf{N}_x(\mathbf{x}, \epsilon_x)$) and the desired output should be within a range (with respect to diss$_y$) with probability $P$ where outputs are analyzed in $l$ groups (left). A particular case where the dissimilarity functions are $\ell_2$ norm and $l = t/2$ (right).

Figure 1 (left) illustrates the schematic of certified robustness for regression. As underpinned by Eq. (2), we are interested in finding the tightest $\epsilon_x$ where all samples within the region defined around **x**, are allowed to push on average only $1 - P$ fraction of generated samples beyond the accepted region of the output. In general cases, outputs can be grouped in $l$ categories to perform the analysis, however, for simplicity within this paper we consider $l = t$.

## 4.1 Robustness of Base Regression Models

In this section, we analyze the probabilistic robustness of an arbitrary model $\mathbf{f}_\theta(\mathbf{x})$, that has not undergone randomized smoothing. In the next section, we will introduce randomized smoothing for regression, and analyze its impact on robustness. See Appendix A for a proof of the following result.

**Theorem 1:** (Certification of General Models Against $\ell_2$-Bounded Attack). *Let $\boldsymbol{f}_\theta(\boldsymbol{x}) : \mathbb{R}^d \to \mathbb{R}^t$ be a (possibly) randomized base regressor and $\boldsymbol{e} \sim \mathcal{N}(\boldsymbol{0}, \sigma^2 \boldsymbol{I})$. Suppose for some example $(\boldsymbol{x}, \boldsymbol{y})$,*

$$\mathbb{P}\{\mathrm{diss}_y(\mathbf{f}_\theta(\mathbf{x} + \mathbf{e})_i, \mathbf{y}_i) \leq \epsilon_{y_i}\} \geq \underline{p_{A_i}}, \forall i \in [t] \tag{3}$$

*where $\underline{p_{A_i}}$ is the lower bound on the probability of accepting prediction in the $i^{th}$ output variable. Then referring to Definition 1, $\boldsymbol{f}_\theta(\boldsymbol{x} + \boldsymbol{e})$ is probabilistic certifiably robust at example $(\boldsymbol{x}, \boldsymbol{y})$, for a $\ell_2$-norm dissimilarity in the input, chosen probability $P \leq \min_{i \in [t]} \underline{p_{A_i}}$, output radii $\epsilon_{y_1}, \ldots, \epsilon_{y_t}$ and input radius*

$$\epsilon_x = \min_{i \in [t]} \sigma \big( \Phi^{-1}(\underline{p_{A_i}}) - \Phi^{-1}(P) \big). \tag{4}$$

Several observations can be made from Theorem 1.

**General output dissimilarities.** This result is general and it is valid for any output vicinity (either diss$_y(\cdot, \cdot)$ or $\epsilon_y$) that the user is interested in since they indirectly affect the results through changes in $\underline{p_{A_i}}$, not in the closed-form expression for the certified radius.

**Interplay between $\underline{p_{A_i}}$'s and $\sigma$.** Theorem 1 is only valid when the user-chosen probability $P \leq \underline{p_{A_i}}, \forall i \in [t]$, otherwise, it returns $\epsilon_x \leq 0$. This observation makes sense because if $\underline{p_{A_i}} < P$, it means that without any adversarial perturbation, the predicted result is not already in favor of the user. Therefore, additional perturbation of the input does not lead to a better performance. One way to satisfy this condition is to use smaller $\sigma$ (sacrificing the tightness of the certification bound).

**Assumption free on base regressor.** Similar to the results for classification tasks (Cohen et al., 2019), Theorem 1 does not make any assumption about $\mathbf{f}_\theta(.)$ and only requires black-box access to compute $\underline{p_{A_i}}$.

**Interplay between $\epsilon_x$ and other parameters.** The certificate radius $\epsilon_x$ is large when (i) the noise level of the isotropic Gaussian smoothing ($\sigma$) is large, (ii) $\underline{p_A}$ is large (ideal when $\underline{p_A}_i \to 1$ meaning that the evaluated **x** is significantly stable), (iii) user is happy with smaller $P$ values which gives a better margin to change the input, or finally (iv) having almost the same sensitivity in all output variables to push $\min_{i \in [t]} \epsilon_{xi}$ towards larger values.

**Abstention.** Theorem 1 has the flexibility to abstain from certifying some of the output variables (by excluding them from the $[t]$ or using larger $\epsilon_y$s) and offer a wider certified radius of the input.

**Estimating $\underline{p_A}_i$'s in practice.** The result in Theorem 1 relies on the exact $\underline{p_A}$, while in reality, this value is not exact unless $n \to \infty$ (large sample regime). Here, we suggest using the confidence interval provided by Clopper & Pearson (1934). Our estimator of $p_A$ follows a Binomial distribution and it is known that for such a parameter the lower bound $\underline{p_A}$ containing the true parameter $p^*$ with confidence level $1 - \frac{\alpha}{2}$, satisfies the equality $\sum_{k=X}^{n} \binom{n}{k} \underline{p_A}^k (1 - \underline{p_A})^{n-k} = \frac{\alpha}{2}$. This estimation is exact and $X$ is the number of successes observed in the sample set containing $n$ samples. See more in Appendix B.

# 5 Randomized Smoothing for Regression

RS for classification tasks is defined as $g(\mathbf{x}) = \arg\max_{c \in \mathbf{Y}} \mathbb{P}(f(\mathbf{x} + \mathbf{e}) = c)$, $\mathbf{e} \sim \mathcal{N}(\mathbf{0}, \sigma^2 \mathbf{I})$ where $\mathbf{Y}$ includes all the class labels. As this smoothing technique integrates mass of votes made for each class, it is not feasible to be used for regression since regression problems deal with continuous output variables. Therefore, we use the averaging function i.e., $\mathbf{g}(\mathbf{x}) = \mathbb{E}\{\mathbf{f}_\theta(\mathbf{x} + \mathbf{e})\}$, $\mathbf{e} \sim \mathcal{N}(\mathbf{0}, \sigma^2 \mathbf{I})$.

## 5.1 Certifying Randomized Smoothing: Asymptotic Case

We have explored the robustness of base $\mathbf{f}_\theta(.)$ through Theorem 1. It is natural to ask: Can smoothing by averaging improve the robustness of the base regression model? If yes, how much improvement can be obtained and what is the behavior of averaging for a large sample regime ($n \to \infty$)? In contrast to the discrete cases, where majority voting can tolerate votes against the target class if they are not in the majority, in the case of smoothing using averaging, intuition suggest that even a single outlier outcome can drastically shift the mean. This behavior is known as the zero breakdown point of the sample average in robust statistics (Rekavandi et al., 2021). In response to these challenges, we first find the approximate probability of returning an acceptable outcome using the average function in the case that $n \to \infty$. See Appendix C for a proof of the following result.

**Theorem 2:** (Asymptotic Behaviour of $\mathbf{g}(\mathbf{x})$ Against a Fixed Attack). *Let $\boldsymbol{f}_\theta(\boldsymbol{x}) : \mathbb{R}^d \to \mathbb{R}^t$ be a (possibly) randomized base regressor and suppose for a given $\boldsymbol{x}$ and $\boldsymbol{\delta}$, outputs generated by $\boldsymbol{f}_\theta(\boldsymbol{x} + \boldsymbol{\delta} + \boldsymbol{e})$, with $\boldsymbol{e} \sim \mathcal{N}(\boldsymbol{0}, \sigma^2 \boldsymbol{I})$ are independent and identically distributed with unknown mean $\boldsymbol{m} \in \mathbb{R}^t$ and unknown bounded covariance $\boldsymbol{\Sigma} \in \mathbb{R}^{t \times t}$. If the accepted region (set) for each output target variable is convex then for the user-defined $\boldsymbol{\epsilon}_y$, as $n \to \infty$, $\mathbb{P}\{\boldsymbol{g}_n(\boldsymbol{x} + \boldsymbol{\delta}) \in \prod_{i=1}^{t} \mathbf{N}_y(\boldsymbol{y}_i, \epsilon_{y_i})\}$ is given by*

$$\boldsymbol{\Phi}\left(\sqrt{n}\hat{\boldsymbol{\Sigma}}^{-\frac{1}{2}}(\mathbf{u}_b - \mathbf{g}_n(\mathbf{x} + \boldsymbol{\delta}))\right) - \sum_{k=1}^{t}(-1)^{k-1} \sum_{\mathcal{D} \in \mathcal{R}_k} \boldsymbol{\Phi}\left(\sqrt{n}\hat{\boldsymbol{\Sigma}}^{-\frac{1}{2}}(\mathbf{c}_{\mathcal{D}} - \mathbf{g}_n(\mathbf{x} + \boldsymbol{\delta}))\right), \tag{5}$$

*where $\boldsymbol{\Phi}(\cdot)$ is the cumulative distribution function of a standard multivariate normal distribution, $\boldsymbol{c}_\mathcal{D}$ uses the lower bounds $\boldsymbol{l}_{b_i}$ for all $i \in \mathcal{D}$ and the upper bounds $\boldsymbol{u}_{b_i}$ for all $i \notin \mathcal{D}$ such that $\mathcal{R}_k$ denotes the class of all subsets of $[t]$ with exactly $k$ elements, and $\boldsymbol{g}_n(\boldsymbol{x})$ is given by*

$$\mathbf{g}_n(\mathbf{x}) = \frac{1}{n}\sum_{i=1}^{n} \mathbf{f}_\theta(\mathbf{x} + \mathbf{e}_i). \tag{6}$$

*In the above, $\boldsymbol{u}_b$ and $\boldsymbol{l}_b$ are upper and lower bounds on the accepted region in output, and $\hat{\boldsymbol{\Sigma}}$ is a consistent covariance estimator.*

**Example 1:** For clarity and better interoperability of the results in Theorem 2, let's consider the case where $t = 2$, representing a regression model with two output variables. Then the quantity in (5) reduces to $\boldsymbol{\Phi}(\sqrt{n}\hat{\boldsymbol{\Sigma}}^{-\frac{1}{2}}(\mathbf{u}_b - \mathbf{g}_n(\mathbf{x} + \boldsymbol{\delta}))) - \boldsymbol{\Phi}(\sqrt{n}\hat{\boldsymbol{\Sigma}}^{-\frac{1}{2}}([l_{b_1}, u_{b_2}]^\top - \mathbf{g}_n(\mathbf{x} + \boldsymbol{\delta}))) - \boldsymbol{\Phi}(\sqrt{n}\hat{\boldsymbol{\Sigma}}^{-\frac{1}{2}}([u_{b_1}, l_{b_2}]^\top - \mathbf{g}_n(\mathbf{x} + \boldsymbol{\delta}))) + \boldsymbol{\Phi}(\sqrt{n}\hat{\boldsymbol{\Sigma}}^{-\frac{1}{2}}(\mathbf{l}_b - \mathbf{g}_n(\mathbf{x} + \boldsymbol{\delta})))$. As can be observed, the first summation in (5) only takes care of signs and the second summation constructs the terms based on lower/upper bounds formed by $\mathbf{c}_\mathcal{D}$, to accurately compute the probability without omissions or redundant calculations of any region.

**Remark 2:** Based on results in Theorem 2, if $\mathbf{m}$ falls in the accepted region of the output, we can conclude as $n \to \infty$, with a high probability $\mathbf{g}_n(\mathbf{x} + \boldsymbol{\delta})$ will stay in the accepted region. This helps to increase the probability of output robustness from $p$ (robustness probability for base regression, i.e., $n = 1$) to $1 - \xi$ (sufficiently small $\xi$, i.e., $0 < \xi \ll 1$).

**Remark 3:** Theorem 2 claims an interesting result for the large sample regime. However, both $\mathbf{m}$ and $\boldsymbol{\Sigma}$ are unknown and they are both functions of $\boldsymbol{\delta}$ that is unknown in practice. Therefore, how the result can be interpreted to determine the bound for the general form of $\boldsymbol{\delta}$ is ambiguous. In other words, if a user is interested in setting the value of Eq. (5) to a certain value, there is no way to find the constraint on $\boldsymbol{\delta}$. Therefore, we need more assumptions to establish a connection between the input and output.

## 5.2 Certifying Randomized Smoothing: Asymptotic, Bounded Output Case

Remark 3 demonstrates that further assumptions are needed for practical certificates. One assumption that can involve $\boldsymbol{\delta}$ in the output probability is considering a range for output variables, i.e., component-wise box constraints $\mathbf{l} \le \mathbf{f}_\theta(\mathbf{z}) \le \mathbf{u}, \forall \mathbf{z}$ where $\mathbf{l}$ and $\mathbf{u}$ construct a sufficiently large region. We argue that this assumption is relatively weak: it is common in learning theory to assume bounded outputs, outputs can be artificially bounded through clipping, and in most applications, users estimate continuous variables such as location in a local area, angle, income, etc. which are all bounded naturally. We consider the robustness property of the averaging function for this class of models. See Appendix D for a proof.

**Theorem 3:** (Certification of $\mathbf{g}(\mathbf{x})$ Against $\ell_2$ Attack for Bounded Outputs). *Let $\boldsymbol{f}_\theta(\boldsymbol{x}) : \mathbb{R}^d \to \mathbb{R}^t$ be a (possibly) randomized base regressor and suppose outputs generated by $\boldsymbol{f}_\theta(\boldsymbol{x})$ are independent and identically distributed with a user-defined $\boldsymbol{\epsilon}_y$ (equivalent to $\boldsymbol{u}_b$ and $\boldsymbol{l}_b$) to define the accepted region. Suppose for $\boldsymbol{e} \sim \mathcal{N}(\boldsymbol{0}, \sigma^2 \boldsymbol{I})$, $\forall \|\boldsymbol{\delta}\|_2 \le \epsilon_x$, and an arbitrary value of $p$, as stated in (4), $\forall i \in [t]$:*

$$\mathbb{P}\{ diss_y(\boldsymbol{f}_\theta(\boldsymbol{x} + \boldsymbol{\delta} + \boldsymbol{e})_i, \boldsymbol{y}_i) \le \epsilon_{y_i} \} \ge p. \tag{7}$$

*Considering bounded output cases, i.e., $\boldsymbol{l} \le \boldsymbol{f}_\theta(\boldsymbol{z}) \le \boldsymbol{u}, \forall \boldsymbol{z}$ subject to $\boldsymbol{l} \le \boldsymbol{l}_b \le \boldsymbol{u}_b \le \boldsymbol{u}$, and if for those samples which are accepted by the user, we have $|\mathbb{E}\{\boldsymbol{f}_\theta(\boldsymbol{x} + \boldsymbol{\delta} + \boldsymbol{e})\}_i - \boldsymbol{f}_\theta(\boldsymbol{x})_i| \le \tau, 0 \le \tau \le \min(\boldsymbol{f}_\theta(\boldsymbol{x}) - \boldsymbol{l}_b, \boldsymbol{u}_b - \boldsymbol{f}_\theta(\boldsymbol{x})),$[1] for the convex accepted region (set) and as $n \to \infty$, the following inequality holds $\forall i \in [t]$*

$$\mathbb{P}\{ diss_y(\boldsymbol{g}_n(\boldsymbol{x} + \boldsymbol{\delta})_i, \boldsymbol{y}_i) \le \boldsymbol{\epsilon}_{y_i} \} \ge$$
$$\min_{i \in [t]} \begin{cases} I_p(\lceil n(1 - \frac{\boldsymbol{u}_{b_i} - \boldsymbol{f}_\theta(\boldsymbol{x})_i - \tau}{\boldsymbol{u}_i - \boldsymbol{f}_\theta(\boldsymbol{x})_i - \tau}) \rceil, \lceil n \frac{\boldsymbol{u}_{b_i} - \boldsymbol{f}_\theta(\boldsymbol{x})_i - \tau}{\boldsymbol{u}_i - \boldsymbol{f}_\theta(\boldsymbol{x})_i - \tau} \rceil + 1), & if \ \frac{\boldsymbol{u}_i - \boldsymbol{u}_{b_i}}{\boldsymbol{u}_i - \boldsymbol{f}_\theta(\boldsymbol{x})_i - \tau} \ge \frac{\boldsymbol{l}_i - \boldsymbol{l}_{b_i}}{\boldsymbol{f}_\theta(\boldsymbol{x})_i - \tau - \boldsymbol{l}_i} \\ I_p(\lceil n(1 - \frac{\boldsymbol{f}_\theta(\boldsymbol{x})_i - \tau - \boldsymbol{l}_{b_i}}{\boldsymbol{f}_\theta(\boldsymbol{x})_i - \tau - \boldsymbol{l}_i}) \rceil, \lceil n(\frac{\boldsymbol{f}_\theta(\boldsymbol{x})_i - \tau - \boldsymbol{l}_{b_i}}{\boldsymbol{f}_\theta(\boldsymbol{x})_i - \tau - \boldsymbol{l}_i}) \rceil + 1), & otherwise, \end{cases} \tag{8}$$

*where $I_p(a, b)$ is the regularized incomplete beta function defined as $I_p(a, b) = \frac{1}{B(a,b)} \int_0^p t^{a-1}(1-t)^{b-1}dt$ and $B(a, b)$ is the complete beta function.*

**Remark 4:** As the accepted region bounds, i.e., $\mathbf{u}_b$ and $\mathbf{l}_b$ get tighter around $\mathbf{f}_\theta(\mathbf{x})$, the lower bound of Theorem 3 tends to 0. In other words, the user does not tolerate any variation in the output, therefore, the

---

[1]Although this assumption may look strange, asymptotically and when the accepted region is symmetric, this assumption makes more sense because the range is small and the distribution in that range can be approximated by a symmetric distribution. Then the best representative of the range is the average itself, and for the worst-case scenario, we use $\mathbf{f}_\theta(\mathbf{x}) \pm \tau$.

method does not guarantee anything after taking the average.

**Remark 5:** Fixing the number of samples and the lower/upper bounds of the output in Theorem 3, the lower bound of $\mathbb{P}\{\text{diss}_y(\mathbf{g}_n(\mathbf{x}+\boldsymbol{\delta})_i, \mathbf{y}_i) \leq \boldsymbol{\epsilon}_{y_i}\}$ is monotonically increasing with $p$ (therefore, the inverse exists, i.e., $I^{-1}(.; a, b)$ exists), and as $p \to 1$, $\mathbb{P}\{\text{diss}_y(\mathbf{g}_n(\mathbf{x}+\boldsymbol{\delta})_i, \mathbf{y}_i) \leq \boldsymbol{\epsilon}_{y_i}\} \to 1$.

Based on the above results for a bounded output, if one is interested in finding the upper bound on $\|\boldsymbol{\delta}\|_2$ to ensure the average value is valid with probability $1/2 \leq q \leq 1$, it is suggested to first estimate $\hat{p}_i$ for $i \in [t]$ by

$$\hat{p}_i = \begin{cases} I^{-1}(q; \lceil n(1 - \frac{\mathbf{u}_{b_i} - \mathbf{f}_\theta(\mathbf{x})_i - \tau}{\mathbf{u}_i - \mathbf{f}_\theta(\mathbf{x})_i - \tau}) \rceil, \lceil n\frac{\mathbf{u}_{b_i} - \mathbf{f}_\theta(\mathbf{x})_i - \tau}{\mathbf{u}_i - \mathbf{f}_\theta(\mathbf{x})_i - \tau} \rceil + 1), & \text{if } \frac{\mathbf{u}_i - \mathbf{u}_{b_i}}{\mathbf{u}_i - \mathbf{f}_\theta(\mathbf{x})_i - \tau} \geq \frac{\mathbf{l}_i - \mathbf{l}_{b_i}}{\mathbf{f}_\theta(\mathbf{x})_i - \tau - \mathbf{l}_i} \\ I^{-1}(q; \lceil n(1 - \frac{\mathbf{f}_\theta(\mathbf{x})_i - \tau - \mathbf{l}_{b_i}}{\mathbf{f}_\theta(\mathbf{x})_i - \tau - \mathbf{l}_i}) \rceil, \lceil n(\frac{\mathbf{f}_\theta(\mathbf{x})_i - \tau - \mathbf{l}_{b_i}}{\mathbf{f}_\theta(\mathbf{x})_i - \tau - \mathbf{l}_i}) \rceil + 1), & \text{otherwise,} \end{cases} \tag{9}$$

in a reverse process (as we know $I^{-1}(.; a, b)$ exists), and then apply Theorem 1 to find $\epsilon_x$ for $\mathbf{g}_n(\mathbf{x})$, i.e.,

$$\epsilon_x = \min_{i \in [t]} \sigma\big(\Phi^{-1}(\underline{p_{A}}_i) - \Phi^{-1}(\hat{p}_i)\big). \tag{10}$$

### 5.3 Certifying Randomized Smoothing: Non-Asymptotic Case

To provide an analytical result for the finite sample scenario i.e., $n < \infty$, we introduce a framework we call *Discounted Certificate*. Although for the base regression model, we consider the $\mathbf{l}_b$ and $\mathbf{u}_b$ as the accepted region boundaries, for $\mathbf{g}_n(\mathbf{x})$ the user is asked to apply some discount factor $\beta \geq 0$ to make the accepted range wider. By doing so, we can now consider the worst-case scenario (putting all the accepted samples in the boundary instead of placement at $\mathbf{f}_\theta(\mathbf{x}) \pm \tau$) and leverage this additional margin added by the user. In other words, the new discounted upper and lower bounds of the accepted region are $\mathbf{u}_b + \beta|\mathbf{u}_b - \mathbf{f}_\theta(\mathbf{x})|$ and $\mathbf{l}_b - \beta|\mathbf{l}_b - \mathbf{f}_\theta(\mathbf{x})|$. A side benefit of this approach is that we relax the Theorem 3 assumption that $|\mathbb{E}\{\mathbf{f}_\theta(\mathbf{x}+\boldsymbol{\delta}+\mathbf{e})\}_i - \mathbf{f}_\theta(\mathbf{x})_i| \leq \tau$. See Appendix E for a proof.

**Proposition 1:** (Discounted Certification of $\mathbf{g}(\mathbf{x})$ against $\ell_2$ Attack for Bounded Outputs). *Let $\boldsymbol{f}_\theta(\boldsymbol{x}) : \mathbb{R}^d \to \mathbb{R}^t$ be a (possibly) randomized base regressor and suppose outputs generated by $\boldsymbol{f}_\theta(\boldsymbol{x})$ are independent and identically distributed with a user-defined $\boldsymbol{\epsilon}_y$ (equivalent to $\boldsymbol{u}_b$ and $\boldsymbol{l}_b$) to define the accepted region. Suppose for $\boldsymbol{e} \sim \mathcal{N}(\boldsymbol{0}, \sigma^2 \boldsymbol{I})$, $\forall\|\boldsymbol{\delta}\|_2 \leq \epsilon_x$, and an arbitrary value of $p$, as stated in (4), $\forall i \in [t]$:*

$$\mathbb{P}\{diss_y(\boldsymbol{f}_\theta(\boldsymbol{x}+\boldsymbol{\delta}+\boldsymbol{e})_i, \boldsymbol{y}_i) \leq \boldsymbol{\epsilon}_{y_i}\} \geq p. \tag{11}$$

*Considering bounded output cases, i.e., $\boldsymbol{l} \leq \boldsymbol{f}_\theta(\boldsymbol{z}) \leq \boldsymbol{u}, \forall \boldsymbol{z}$ subject to $\boldsymbol{l} \leq \boldsymbol{l}_b \leq \boldsymbol{u}_b \leq \boldsymbol{u}$ and assuming the accepted region (set) for each output target variable to be convex, then given a discount factor $\beta \geq 0$ such that $\boldsymbol{l} \leq \boldsymbol{l}_b - \beta|\boldsymbol{f}_\theta(\boldsymbol{x}) - \boldsymbol{l}_b| \leq \boldsymbol{u}_b + \beta|\boldsymbol{f}_\theta(\boldsymbol{x}) - \boldsymbol{u}_b| \leq \boldsymbol{u}$ holds, then the following inequality holds for $\forall i \in [t]$*

$$\mathbb{P}\{diss_y(\boldsymbol{g}_n(\boldsymbol{x}+\boldsymbol{\delta})_i, \boldsymbol{y}_i) \leq (1+\beta)\boldsymbol{\epsilon}_{y_i}\} \geq$$
$$\min_{i \in [t]} \begin{cases} I_p(\lceil n(1 - \frac{\beta|\boldsymbol{u}_{b_i} - \boldsymbol{f}_\theta(\boldsymbol{x})_i|}{\boldsymbol{u}_i - \boldsymbol{u}_{b_i}}) \rceil, \lceil n(\frac{\beta|\boldsymbol{u}_{b_i} - \boldsymbol{f}_\theta(\boldsymbol{x})_i|}{\boldsymbol{u}_i - \boldsymbol{u}_{b_i}}) \rceil + 1), & \text{if } \frac{|\boldsymbol{u}_{b_i} - \boldsymbol{f}_\theta(\boldsymbol{x})_i|}{\boldsymbol{u}_i - \boldsymbol{u}_{b_i}} \leq \frac{|\boldsymbol{l}_{b_i} - \boldsymbol{f}_\theta(\boldsymbol{x})_i|}{\boldsymbol{l}_{b_i} - \boldsymbol{l}_i} \\ I_p(\lceil n(1 - \frac{\beta|\boldsymbol{l}_{b_i} - \boldsymbol{f}_\theta(\boldsymbol{x})_i|}{\boldsymbol{l}_{b_i} - \boldsymbol{l}_i}) \rceil, \lceil n\frac{\beta|\boldsymbol{l}_{b_i} - \boldsymbol{f}_\theta(\boldsymbol{x})_i|}{\boldsymbol{l}_{b_i} - \boldsymbol{l}_i} \rceil + 1), & \text{otherwise,} \end{cases} \tag{12}$$

*where $I_p(a, b)$ is the regularized incomplete beta function defined as $I_p(a, b) = \frac{1}{B(a,b)} \int_0^p t^{a-1}(1-t)^{b-1} dt$ and $B(a, b)$ is the complete beta function.*

As outlined in Proposition 1, the user is required to incorporate a discount factor, represented by a positive constant $\beta$, into the acceptable output range to establish a certificate for the worst-case scenario. As a consistency check, when $\beta \to 0$ in (12), the lower bound approaches zero. This implies that in the extreme case where all valid output samples are already at the boundary, a single invalid output can push the average beyond the validity threshold. Since $\beta \to 0$, eliminates any discount or margin, there is no justification for the result, leading the lower bound on the probability to converge to zero.

A drawback of this approach is that if the user chooses not to apply such a discount, no certification can be

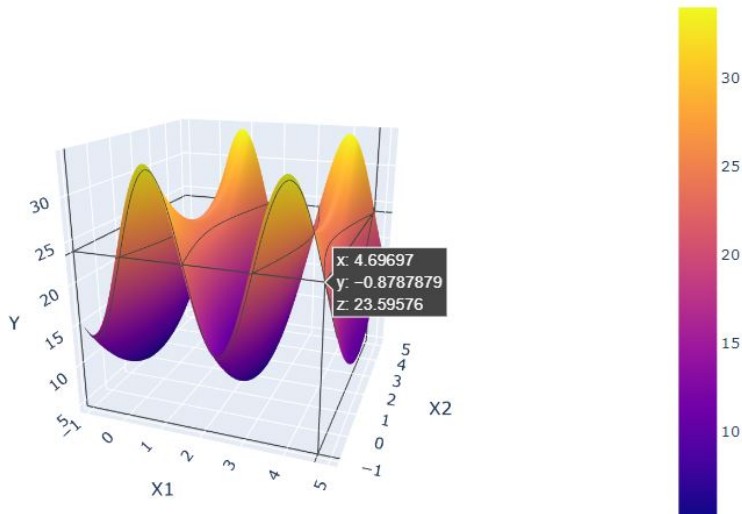

Figure 2: The two-dimensional function used for the simulation.

provided for the result. While this may initially seem discouraging, we propose an alternative strategy. If the user cannot extend the output region, we introduce an artificial (conservative) marginal zone by adjusting $\epsilon_y$ thereby reducing the acceptable region for estimating $p_{A_i}$ for $\forall i \in [t]$. Although this adjustment may yield a weaker certificate compared to the one for $\mathbf{f}_\theta(\mathbf{x})$, it ensures that the certification is not entirely lost and that any obtained certificates remain valid.

### 5.4  Adversarial Training

To obtain improved tightness from randomized smoothing, it is first required to have a robust base regressor as stated in Theorem 1. Cohen et al. (2019) argued that to increase the robustness of the base classifier—if retraining of the network is permitted—one can use Gaussian noise augmentation and train the network. Although our proposed method is designed for the regression task, it can leverage the same concept. In other words, we can use $\mathbf{x} + \mathbf{e}$, $\mathbf{e} \sim \mathcal{N}(\mathbf{0}, \sigma^2 \mathbf{I})$ to train the base regression model with the same ground truth $\mathbf{y}$. In the case of black-box access, other strategies such as denoisers attached to the base regression—e.g., what has been proposed in Salman et al. (2020); Carlini et al. (2023)—can be utilized.

## 6  Experimental Results

In this section, we perform experiments to empirically validate our theoretical results. For synthetic simulations, we present the results for an example function that demonstrates sharp variations in output. We then apply the proposed methods on a camera re-localization task (Rekavandi et al., 2023) based on images. All simulations and experiments were conducted using an Intel(R) Core(TM) i7-9750H CPU running at 2.60GHz (with a base clock speed of 2.59GHz) and 16GB of RAM.

**Simulation study.** We begin the simulation with an example base model $f : \mathbb{R}^2 \rightarrow \mathbb{R}$ given by

$$f(\mathbf{x}) = 10 \sin(2x_1) + (x_2 - 2)^2 + 15. \tag{13}$$

This function was investigated for the interval $-1 < x_1, x_2 < 5$ with Figure 2 illustrating this sharp function for the defined range. As all the results in this paper are point-wise certificates, we only selected the integer points in these ranges to derive the certificate radii. We set $\sigma = 0.23$, $\epsilon_y = 6$ for the $\ell_1$ output norm, $U = 35$, $L = 0$, $\tau = 0$, $n = 10K$, to ensure that the user-defined probability $P = 80\%$ is always less than $p_A$. As $n$ is large, we used the estimated $p_A$ as the $\underline{p_A}$ and skipped the use of the Clopper-Pearson lower bound estimator (see Appendix B). We also selected $\overline{\beta} \in \{1.5, 2\}$ for the discounted certification algorithm. Figure 3 visualizes the theoretical certificate radii derived from Eq. (4) for $f(\mathbf{x})$ (blue), inequality (8) for $g(\mathbf{x})$ (red),

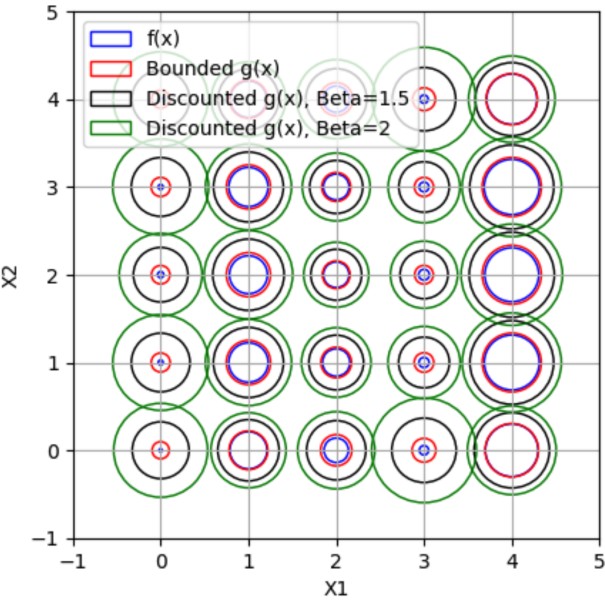

Figure 3: Theoretical certificates derived for $f(\mathbf{x})$ (blue), $g(\mathbf{x})$ for well-behaved base regression (red), and discounted certificates of $g(\mathbf{x})$ (black and green) where the user set $P = 80\%$ (right).

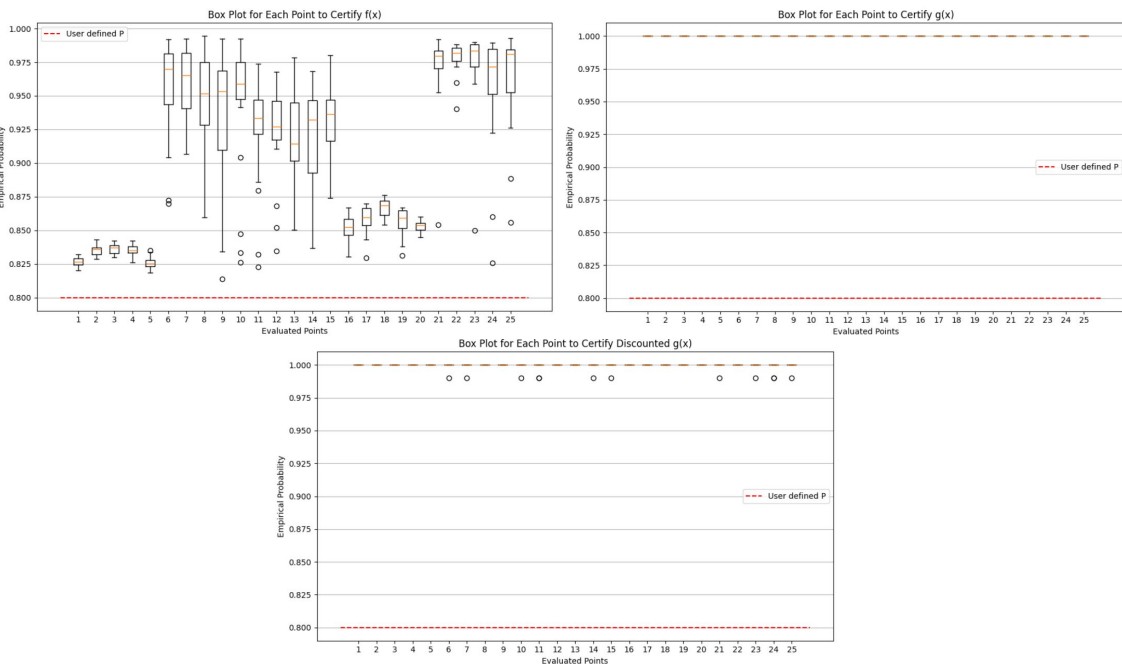

Figure 4: Empirical probability of valid output in comparison with desired probability defined by the user (80%) for $f(\mathbf{x})$ (top left), $g(\mathbf{x})$ (top right), discounted $g(\mathbf{x})$ (bottom).

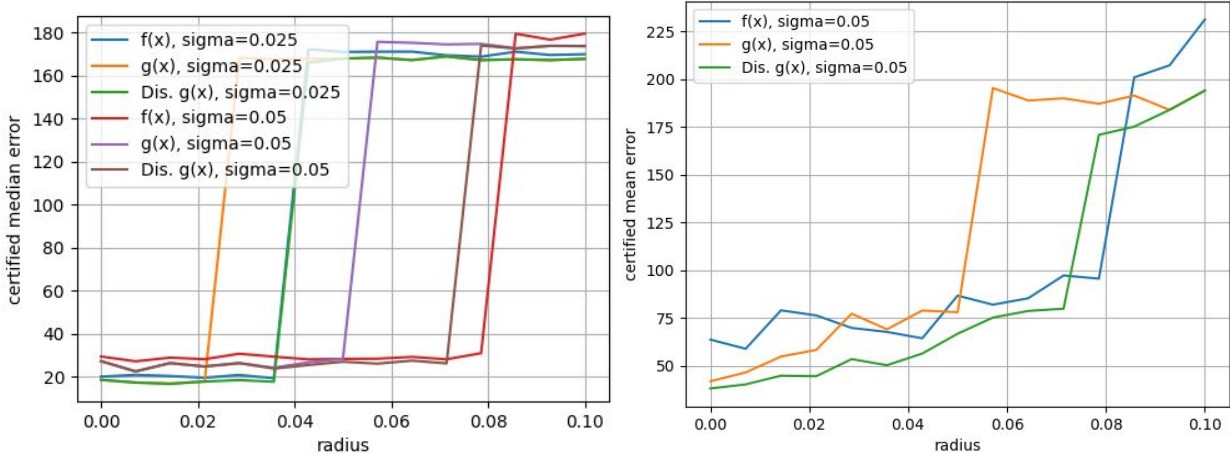

Figure 5: Certified median (left) and mean (right) error in DSAC$^*$ as a function of $r$.

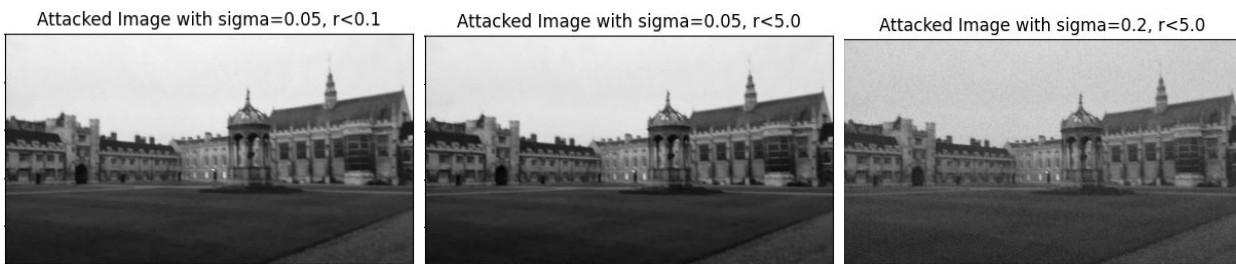

Figure 6: Examples of adversarial images contaminated with noise. As shown the changes in the certified range and considered noise are not visible to the human vision system, unless they become significant in magnitude.

and inequality (12) for $g(\mathbf{x})$ where the output validity range is discounted in two different rates (black and green).

The certified radius for $f(\mathbf{x})$ is wider for the points that return smoother changes in the output and is small for sharper regions. Randomized smoothing in Theorem 3, offers a slightly better certificate compared to the certification of $f(\mathbf{x})$, however, this is true only for those points that satisfy the assumed condition. This improvement is small mainly due to a wide range of output variability and sensitivity of averaging function to a single largely deviated point. Finally, for discounted certification of $g(\mathbf{x})$, considering $\beta = 1.5$, we attained a better certificate (black) than any other approaches. This certificate becomes even better when a larger discount is applied (green for $\beta = 2$). Note that all of these certificates become smaller and smaller when the user-defined desired robustness probability increases (see Appendix F). Figure 4 visualizes the empirical probabilities obtained over 25 points evaluated for $f(\mathbf{x})$ and $g(\mathbf{x})$, respectively for 20 trials. As shown in Figure 4 (top left), the proposed radii for the evaluated points (in $f(\mathbf{x})$) perfectly satisfy the user-defined probability and they are all uniformly above this threshold as expected from the results in Theorem 1. As shown, at some points, these empirical gaps are small which empirically shows the tightness of the bound. Figure 4 (top right), shows the results of Theorem 3, and as shown the probabilistic lower bound holds with a great margin though the upper bound on input radius is better than the one for $f(\mathbf{x})$ (red circles in Figure 3). Figure 4 (bottom) illustrates the lower bound and the empirical probabilities for the discounted certificate ($\beta = 1.5$) scenario, and as shown all the empirical values are above the desired level. Although these certificates are sound, they are conservative as expected, since we consider a worst-case scenario in the lower bound derivation.

**Camera re-localization task.** Camera re-localization from a single RGB image is an important task in many applications such as robotic positioning systems, SLAM, autonomous driving, etc. Given an image

in the input, the output of such a system is a multivariate continuous variable and the task is to regress 6 parameters denoted by $\mathbf{p}$, 3 for position and 3 for orientation. Defense against adversarial perturbation for such a system is vital because any wrong predictions in such systems may cause irreparable consequences, e.g., in autonomous driving systems attacks may cause severe accidents with fatalities. DSAC* (Brachmann & Rother, 2022) is a popular technique and adopted in this paper for robustness analysis. Although in the literature, together with median error, the accuracies were reported for some user-defined thresholds, e.g., 0.05% for outdoor scenes, we report the result in terms of median and mean error. The certified median error at radius $r$ in general form is defined as the median of errors computed for each test image as follows:

$$e_K \quad = \quad \text{diss}_y(\mathbf{g}(\mathbf{x} + \boldsymbol{\delta}), \mathbf{p}^*) + \mathbf{1}_{r > \epsilon_x} K, \quad \forall \|\boldsymbol{\delta}\|_2 \leq r$$

where $K$ is an arbitrarily large value and pushes the output error to a larger range if the evaluated radius is beyond the estimated certificate bound. In our simulation, we only consider the position-related parameters in $\mathbf{p}$ and use the reduced form

$$e_K \quad = \quad \|\mathbf{g}(\mathbf{x} + \boldsymbol{\delta}) - \mathbf{p}^*)\|_2 + \mathbf{1}_{r > \epsilon_x} K, \quad \forall \|\boldsymbol{\delta}\|_2 \leq r$$

with $K = 150cm$. For learning of $\underline{p_A}$ using Clopper- Pearson ($\alpha = 0.5$), we used 200 samples and then we used $n = 10$ for each radius to examine models in the Cambridge Great Court scene in the Cambridge Landmarks dataset (Kendall et al., 2015) using the DSAC* pre-trained model. For the Great Court Scene, out of 760 test images, 120 randomly selected images were used (due to the similarity of the images and to reduce the required runtime) to report the certified error rate defined above. We used a threshold of $\epsilon_y = 5m$ for defining the accepted region in the output, with ($U = 85$ and $L = -15$) with $\beta = 2$ and $P = 80\%$. we investigate the range of $r \in [0, 0.1]$ for the scene where the image dimension was $480 \times 854$ pixels. Figure 5 (left) illustrates the certified median error rate using two different smoothing noise variations $\sigma \in \{0.025, 0.05\}$. Generally, as $\sigma$ increases, the error rate also increases, but the certified radius increases. Similar to the classification task (where certified accuracy is used), there is a jump in error rate for $r \geq \epsilon_x$ and this jump occurs in larger radii when $\sigma$ increases. The simple averaging function is beneficial in terms of error rate but less beneficial in terms of offered radii compared to the base regression model. This is mainly due to the large upper/lower bound on the output. Figure 5 (right) on the other hand shows the certified mean error of the pose estimation as a function of $r$, to better illustrate the growth of the error. It can be observed for $r \leq \epsilon_x$, the mean error is growing smoothly and when it reaches this threshold, a significant increase occurs in the error due to activation of the penalty term (+K). However, as shown for both small and large $r$ values, the smoothed functions offer better error rates. Examples of attacked images are shown in Figure 6.

## 7 Conclusion

In this paper, we investigated certified robustness against adversarial perturbation in regression tasks. We analyzed the robustness of a base regression model without any smoothing, using only black-box access to the model. We then proposed a new variant of certification where the outputs are bounded. Subject to the user's flexibility, a discounted certificate approach was proposed for bounded outputs where the result is valid for a finite sample regime. The results were validated using simulation and camera localization, demonstrating effective bounds. A promising direction is to investigate other smoothing functions dedicated to continuous outputs and derivation of probabilistic certified guarantees for unbounded outputs. Although our definition of robustness is general, we only considered the case of $\ell_2$ attacks—results for other threat models would be valuable. In our results, outputs are analyzed individually, while in some applications it might be possible to analyze them by groupings for a joint analysis.

**Broader Impact Statement**

Adversarial examples demonstrate the vulnerability of many machine learning models to manipulation in contested environments. This paper considers defenses (via randomized smoothing) and robustness quantification, which are important approaches to improving resistance to attacks. As such, we believe this work has potential for positive societal benefit.

**Acknowledgement**

This work was supported in part by the Department of Industry, Science, and Resources, Australia under AUSMURI CATCH, and the Australian Research Council under Discovery Project DP220102269.

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

# A   Proof of Base Regression Certification

We repeat the theorem's statement here for convenience, followed by its proof.

**Theorem 1:** (Certification of General Models Against $\ell_2$-Bounded Attack). *Let $\boldsymbol{f}_\theta(\boldsymbol{x}) : \mathbb{R}^d \to \mathbb{R}^t$ be a (possibly) randomized base regressor and $\boldsymbol{e} \sim \mathcal{N}(\boldsymbol{0}, \sigma^2\boldsymbol{I})$. Suppose for some example $(\boldsymbol{x}, \boldsymbol{y})$,*

$$\mathbb{P}\{\mathrm{diss}_y(\mathbf{f}_\theta(\mathbf{x} + \mathbf{e})_i, \mathbf{y}_i) \leq \epsilon_{y_i}\} \geq \underline{p_{A_i}}, \forall i \in [t] \tag{14}$$

*where $\underline{p_{A_i}}$ is the lower bound on the probability of accepting prediction in the $i^{th}$ output variable. Then referring to Definition 1, $\boldsymbol{f}_\theta(\boldsymbol{x} + \boldsymbol{e})$ is probabilistic certifiably robust at example $(\boldsymbol{x}, \boldsymbol{y})$, for a $\ell_2$-norm dissimilarity in the input, chosen probability $P \leq \min_{i \in [t]} \underline{p_{A_i}}$, output radii $\epsilon_{y_1}, \ldots, \epsilon_{y_t}$ and input radius*

$$\epsilon_x = \min_{i \in [t]} \sigma\big(\Phi^{-1}(\underline{p_{A_i}}) - \Phi^{-1}(P)\big). \tag{15}$$

We need the following classical result to prove our theorem.

**Lemma 1:** (Cohen et al., 2019) (Neyman-Pearson for Two Gaussians with Different Means and the Same Variances). *Let $X \sim \mathcal{N}(\boldsymbol{x}, \sigma^2\boldsymbol{I})$ and $Q \sim \mathcal{N}(\boldsymbol{x} + \boldsymbol{\delta}, \sigma^2\boldsymbol{I})$ and let $h : \mathbb{R}^d \to \{0, 1\}$ be a deterministic or random function, then if $S = \{\boldsymbol{z} \in \mathbb{R}^d : \boldsymbol{\delta}^\top \boldsymbol{z} \leq \beta\}$ for some $\beta$ and $\mathbb{P}\{h(X) = 1\} \geq \mathbb{P}\{X \in S\}$, then $\mathbb{P}\{h(Q) = 1\} \geq \mathbb{P}\{Q \in S\}$.*

*Proof of Theorem 1.* The proof of this theorem is analogous to the proof of Theorem 1 of Cohen et al. (2019), since we divide the output space into two regions: acceptable and rejectable zones (like a binary classification task). However, instead of having two classes to compete, the probability score of the regression prediction should beat the user-defined probability level $P$. In this setup, we predict $t$ target variables simultaneously. Without loss of generality, we derive the maximum tolerable deviation in the input for each output variable. Then, based on the definition of the robust regression model, we find the variable with the worst guarantee to find a value valid for all outputs. For $i^{th}$ target variable, let us define the accepted deviation in the output space (by user) as the deviations such that $\mathrm{diss}_y(\mathbf{f}_\theta(\mathbf{x} + \mathbf{e})_i, \mathbf{y}_i) \leq \epsilon_{y_i}$. We denote the probability of this event with $\mathbb{P}\{\mathrm{diss}_y(\mathbf{f}_\theta(\mathbf{x} + \mathbf{e})_i, \mathbf{y}_i) \leq \epsilon_{y_i}\}$. Now let us define random variables $\mathbf{X} = \mathbf{x} + \mathbf{e}$ and $\mathbf{Q} = \mathbf{x} + \boldsymbol{\delta} + \mathbf{e}$, where $\mathbf{e} \sim \mathcal{N}(\mathbf{0}, \sigma^2\mathbf{I})$. $\boldsymbol{\delta}$ represents the adversarial perturbation in the input. Based on the assumption made in the theorem statement, we know that $\mathbb{P}\{\mathrm{diss}_y(\mathbf{f}_\theta(\mathbf{X})_i, \mathbf{y}_i) \leq \epsilon_{y_i}\} \geq \underline{p_{A_i}}$, and the goal is to derive a bound for $\boldsymbol{\delta}$ such that

$$\mathbb{P}\{\mathrm{diss}_y(\mathbf{f}_\theta(\mathbf{Q})_i, \mathbf{y}_i) \leq \epsilon_{y_i}\} \geq P. \tag{16}$$

Defining a half-space $A := \{\mathbf{z} : \boldsymbol{\delta}^\top(\mathbf{z} - \mathbf{x}) \leq \sigma\|\boldsymbol{\delta}\|_2\Phi^{-1}(\underline{p_{A_i}})\}$, we can show that $\mathbb{P}\{\mathbf{X} \in A\} = \underline{p_{A_i}}$, therefore, one can say

$$\mathbb{P}\{\mathrm{diss}_y(\mathbf{f}_\theta(\mathbf{X})_i, \mathbf{y}_i) \leq \epsilon_{y_i}\} \geq \mathbb{P}\{\mathbf{X} \in A\}. \tag{17}$$

Now we apply Lemma 1 with $h(\mathbf{z}) := \mathbf{1}_{\mathrm{diss}_y(\mathbf{f}_\theta(\mathbf{z})_i, \mathbf{y}_i) \leq \epsilon_{y_i}}$ and conclude

$$\mathbb{P}\{\mathrm{diss}_y(\mathbf{f}_\theta(\mathbf{Q})_i, \mathbf{y}_i) \leq \epsilon_{y_i}\} \geq \mathbb{P}\{\mathbf{Q} \in A\}. \tag{18}$$

For $\mathbb{P}\{\mathbf{Q} \in A\}$ we have

$$
\begin{aligned}
&\mathbb{P}\{\mathbf{Q} \in A\} \\
=\ & \mathbb{P}\{\boldsymbol{\delta}^\top(\mathbf{Q} - \mathbf{x}) \leq \sigma\|\boldsymbol{\delta}\|_2\Phi^{-1}(\underline{p_{A_i}})\} \\
=\ & \mathbb{P}\{\sigma\boldsymbol{\delta}^\top\mathcal{N}(\mathbf{0}, \mathbf{I}) \leq \sigma\|\boldsymbol{\delta}\|_2\Phi^{-1}(\underline{p_{A_i}}) - \|\boldsymbol{\delta}\|_2^2\} \\
=\ & \mathbb{P}\left\{\mathcal{N}(\mathbf{0}, \mathbf{I}) \leq \Phi^{-1}(\underline{p_{A_i}}) - \frac{\|\boldsymbol{\delta}\|_2}{\sigma}\right\} \\
=\ & \Phi\left(\Phi^{-1}(\underline{p_{A_i}}) - \frac{\|\boldsymbol{\delta}\|_2}{\sigma}\right).
\end{aligned}
$$

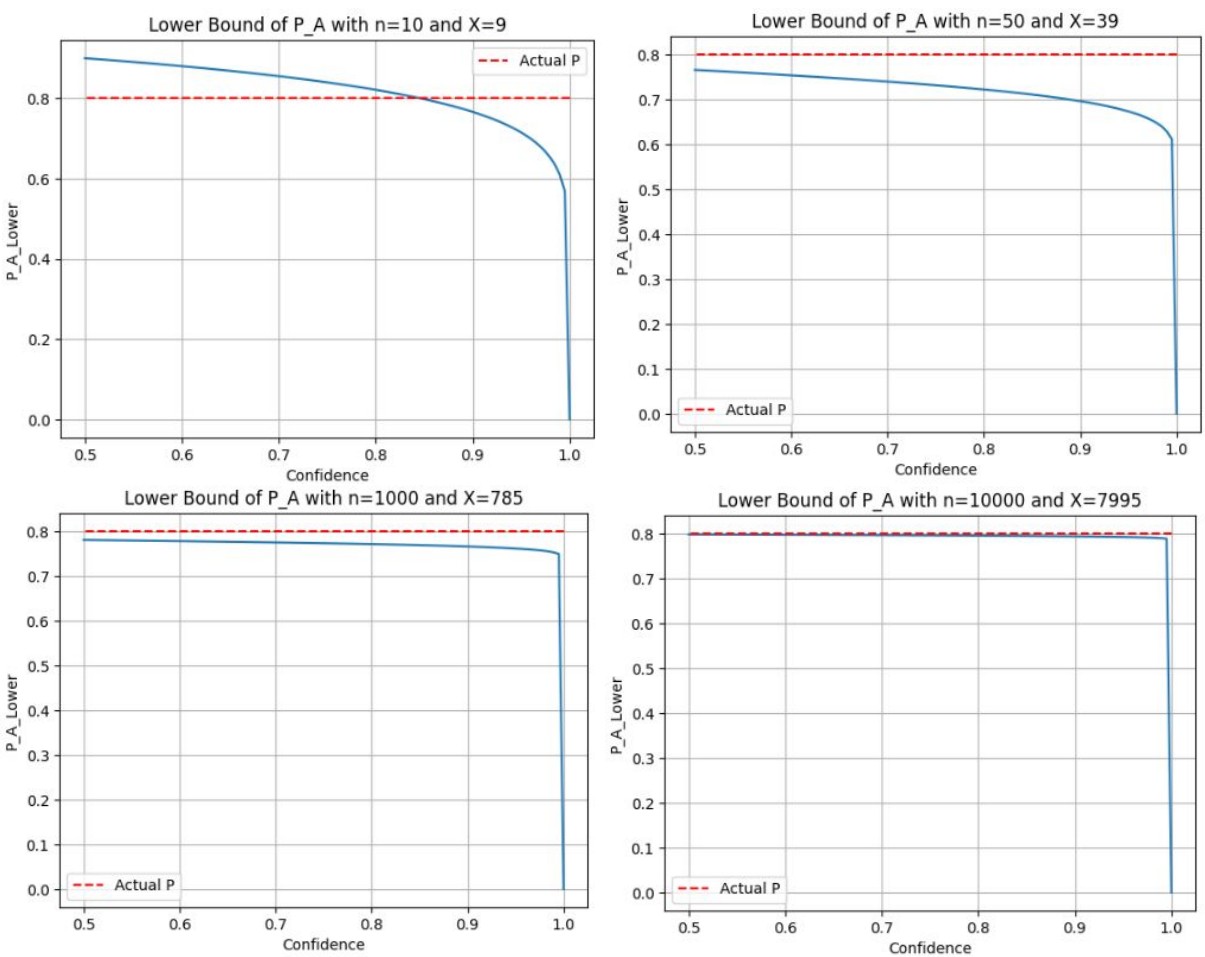

Figure 7: The Clopper-Pearson estimation of $\underline{p_A}$ in different sample sizes and desired confidence levels. Notably, since this is a probabilistic lower bound estimator, as shown in the top-left figure, the actual $p_A$ might also be smaller than the proposed $\underline{p_A}$ when the confidence level is low. However, this is an unlikely event in larger confidence levels and the proposed lower bound is reliable for subsequent analysis. Values of $X$ are randomly drawn from the actual distribution.

Therefore,

$$\mathbb{P}\{\mathrm{diss}_y(\mathbf{f}_\theta(\mathbf{Q})_i, \mathbf{y}_i) \leq \epsilon_{y_i}\} \geq \Phi\left(\Phi^{-1}(\underline{p_{A_i}}) - \frac{\|\boldsymbol{\delta}\|_2}{\sigma}\right).$$

If a user decides to set lower bound of $\mathbb{P}\{\mathrm{diss}_y(\mathbf{f}_\theta(\mathbf{Q})_i, \mathbf{y}_i) \leq \epsilon_{y_i}\}$ to $P$, the corresponding $\boldsymbol{\delta}$ should satisfy

$$\|\boldsymbol{\delta}\|_2 \leq \sigma\big(\Phi^{-1}(\underline{p_{A_i}}) - \Phi^{-1}(P)\big). \tag{19}$$

This $\|\boldsymbol{\delta}\|_2$ indicates the maximum permitted perturbation in the input which guarantees with probability $P$, the $i^{th}$ output is valid to the user. Since we have $t$ output variables, each one with its own permitted perturbation ranges, we estimate $t$ different permitted $\boldsymbol{\delta}$, and finally, give a certificate that all these predictions are certifiable with probability $P$. In other words, we select the worst estimated value given by

$$\epsilon_x = \min_{i \in [t]} \sigma\big(\Phi^{-1}(\underline{p_{A_i}}) - \Phi^{-1}(P)\big). \tag{20}$$

This completes the proof. □

$$\mathbb{P}\{\mathbf{g}_n(\mathbf{x}+\boldsymbol{\delta}) \in \prod_{i=1}^{t} \mathbf{N}_y(\mathbf{y}_i, \epsilon_{y_i})\} = \int \cdots \int_{\mathbf{s} \in \mathbf{N}(\mathbf{z}, \epsilon_y)} \frac{\exp\{-\frac{1}{2}(\mathbf{s} - \mathbf{m})^\top \boldsymbol{\Sigma}^{-1}(\mathbf{s} - \mathbf{m})\}}{\sqrt{(2\pi)^t \det(\boldsymbol{\Sigma})}} d\mathbf{s} \qquad (23)$$

$$\approx \boldsymbol{\Phi}(\sqrt{n}\hat{\boldsymbol{\Sigma}}^{-\frac{1}{2}}(\mathbf{u}_b - \mathbf{g}_n(\mathbf{x}+\boldsymbol{\delta}))) - \sum_{k=1}^{t}(-1)^{k-1} \sum_{\mathcal{D} \in \mathcal{R}_k} \boldsymbol{\Phi}(\sqrt{n}\hat{\boldsymbol{\Sigma}}^{-\frac{1}{2}}(\mathbf{c}_\mathcal{D} - \mathbf{g}_n(\mathbf{x}+\boldsymbol{\delta}))),$$

## B  Estimating $\underline{p_A}$ via Clopper-Pearson

In this section, we numerically show how the Clopper-Pearson (Clopper & Pearson, 1934) interval proposal, i.e., solution of $\sum_{k=X}^{n} \binom{n}{k}\underline{p_A}^k(1 - \underline{p_A})^{n-k} = \frac{\alpha}{2}$ for given $\alpha$, $n$, and $X$ can offer a confidence interval, i.e., $[\underline{p_A}, 1]$ in a finite sample regime with confidence level $1 - \frac{\alpha}{2}$ in containing the actual parameter $p_A^*$. We consider the cases where $n \in \{10, 50, 1000, 10000\}$ and $p_A^*$ is set to $\frac{8}{10}$. For each case, we use a binomial distribution to draw the number of successful events ($\mathbf{X}$), and then use the Clopper-Pearson to estimate $\underline{p_A}$ for various confidence levels. Figure 7 illustrates the lower bounds (in blue) for all these cases and as shown for higher confidence levels, e.g. 0.95, the offered lower bound is uniformly smaller than the actual $p_A$ (shown in red dashed). For larger $n$ values, the gap between $p_A^*$ and $\underline{p_A}$ tends to zero, meaning that for larger $n$, we can simply use the maximum likelihood estimate $\hat{p}_A$ as the $\underline{p_A}$ (as we do in the simulation section). Using this concept, within this paper, the regression certification comes in the following format: "*With confidence $1 - \frac{\alpha}{2}$, the given model at point $(\boldsymbol{x}+\boldsymbol{\delta}, \boldsymbol{y})$, $\forall \|\boldsymbol{\delta}\|_2 \leq \epsilon_x$ is certifiably robust with probability $P$."*

## C  Proof of Asymptotic Certification of Averaging Function

We repeat the theorem's statement here for convenience, followed by its proof.

**Theorem 2:** (Asymptotic Behaviour of $\mathbf{g}(\mathbf{x})$ Against a Fixed Attack). *Let $\boldsymbol{f}_\theta(\boldsymbol{x}) : \mathbb{R}^d \to \mathbb{R}^t$ be a (possibly) randomized base regressor and suppose for a given $\boldsymbol{x}$ and $\boldsymbol{\delta}$, outputs generated by $\boldsymbol{f}_\theta(\boldsymbol{x}+\boldsymbol{\delta}+\boldsymbol{e})$, with $\boldsymbol{e} \sim \mathcal{N}(\boldsymbol{0}, \sigma^2 \boldsymbol{I})$ are independent and identically distributed with unknown mean $\boldsymbol{m} \in \mathbb{R}^t$ and unknown bounded covariance $\boldsymbol{\Sigma} \in \mathbb{R}^{t \times t}$. If the accepted region (set) for each output target variable is convex then for the user-defined $\boldsymbol{\epsilon}_y$, as $n \to \infty$, $\mathbb{P}\{\boldsymbol{g}_n(\boldsymbol{x}+\boldsymbol{\delta}) \in \prod_{i=1}^{t} \mathbf{N}_y(\boldsymbol{y}_i, \epsilon_{y_i})\}$ is given by*

$$\boldsymbol{\Phi}\left(\sqrt{n}\hat{\boldsymbol{\Sigma}}^{-\frac{1}{2}}(\mathbf{u}_b - \mathbf{g}_n(\mathbf{x}+\boldsymbol{\delta}))\right) - \sum_{k=1}^{t}(-1)^{k-1} \sum_{\mathcal{D} \in \mathcal{R}_k} \boldsymbol{\Phi}\left(\sqrt{n}\hat{\boldsymbol{\Sigma}}^{-\frac{1}{2}}(\mathbf{c}_\mathcal{D} - \mathbf{g}_n(\mathbf{x}+\boldsymbol{\delta}))\right), \qquad (21)$$

*where $\boldsymbol{\Phi}(\cdot)$ is the cumulative distribution function of a standard multivariate normal distribution, $\mathbf{c}_\mathcal{D}$ uses the lower bounds $\boldsymbol{l}_{b_i}$ for all $i \in \mathcal{D}$ and the upper bounds $\boldsymbol{u}_{b_i}$ for all $i \notin \mathcal{D}$ such that $\mathcal{R}_k$ denotes the class of all subsets of $[t]$ with exactly $k$ elements, and $\boldsymbol{g}_n(\boldsymbol{x})$ is given by*

$$\mathbf{g}_n(\mathbf{x}) = \frac{1}{n}\sum_{i=1}^{n} \mathbf{f}_\theta(\mathbf{x}+\mathbf{e}_i). \qquad (22)$$

*In the above, $\boldsymbol{u}_b$ and $\boldsymbol{l}_b$ are upper and lower bounds on the accepted region in output, and $\hat{\boldsymbol{\Sigma}}$ is a consistent covariance estimator.*

*Proof.* For clarity of the proof, we refer the reader to Figure 8 which illustrates the setup for the case $(t = d = 2)$. Since $t$-dimensional outputs are i.i.d by assumption, based on the Central Limit Theorem, we have

$$\sqrt{n}\big(\mathbf{g}_n(\mathbf{x}+\boldsymbol{\delta}) - \mathbf{m}\big) \xrightarrow{n \to \infty} \mathcal{N}_t(\mathbf{0}, \boldsymbol{\Sigma}). \qquad (24)$$

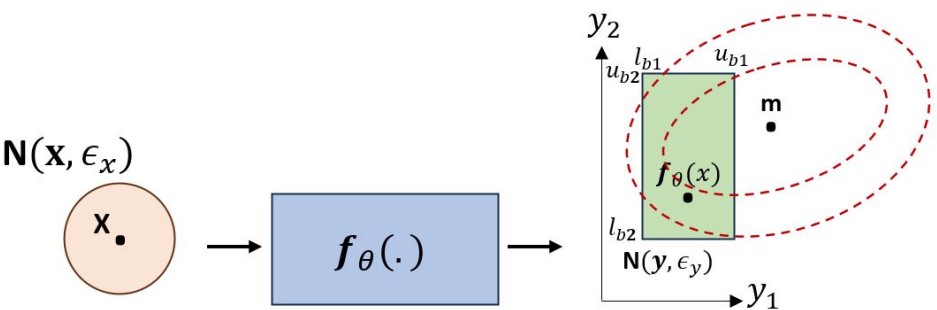

Figure 8: Schematic of the asymptotic behavior of averaging function in output domain for $d = t = 2$. As shown, the $\mathbf{g}_n$ will be Gaussian distributed as $n \to \infty$. The accepted region is shown in green colour.

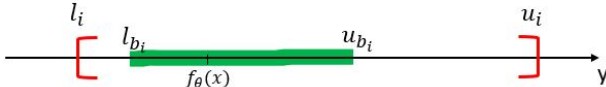

Figure 9: The general schematic of $i^{th}$ output variable when upper and lower bounds of the variable are denoted by $\mathbf{u}_i$ and $\mathbf{l}_i$ and upper and lower bounds of the acceptable region are shown by $\mathbf{u}_{b_i}$ and $\mathbf{l}_{b_i}$. The accepted region is shown in green colour.

In other words, as $n \to \infty$, $\mathbf{g}_n(\mathbf{x} + \boldsymbol{\delta}) \sim \mathcal{N}_t(\mathbf{m}, \frac{1}{n}\boldsymbol{\Sigma})$ which shows the average value concentrates around the expected value of outputs and as $n$ grows, the covariance gets smaller and smaller. Since for a deep network, $\mathbf{m}$ and $\boldsymbol{\Sigma}$ are unknown and they are both functions of $\boldsymbol{\delta}$, by Weak Law of Large Numbers we may replace them with their sample mean and sample covariance estimates or any other consistent estimators. Now, to estimate the probability of $\mathbf{g}_n(\mathbf{x} + \boldsymbol{\delta})$ to be accepted by the user, we need to use (23) which is based on the inclusion-exclusion principle. As is shown this probability function (not a lower bound) is a function of estimated covariance, upper bound and lower bound of the accepted region denoted by $\mathbf{u}_b$ and $\mathbf{l}_b$, $n$ and $\boldsymbol{\delta}$. This completes the proof. $\square$

# D   Proof of Asymptotic Certification of Averaging Function for Bounded Outputs

We repeat the theorem's statement here for convenience, followed by its proof.

**Theorem 3:** (Certification of $\mathbf{g}(\mathbf{x})$ Against $\ell_2$ Attack for Bounded Outputs). *Let $\boldsymbol{f}_\theta(\boldsymbol{x}) : \mathbb{R}^d \to \mathbb{R}^t$ be a deterministic or random base regressor and suppose outputs generated by $\boldsymbol{f}_\theta(\boldsymbol{x})$ are independent and identically distributed with a user-defined $\boldsymbol{\epsilon}_y$ (equivalent to $\boldsymbol{u}_b$ and $\boldsymbol{l}_b$) to define the accepted region. Suppose for $\boldsymbol{e} \sim \mathcal{N}(\boldsymbol{0}, \sigma^2\boldsymbol{I})$, $\forall \|\boldsymbol{\delta}\|_2 \leq \epsilon_x$, and an arbitrary value of $p$, as stated in (4 of the main manuscript), $\forall i \in [t]$:*

$$\mathbb{P}\{diss_y(\boldsymbol{f}_\theta(\boldsymbol{x} + \boldsymbol{\delta} + \boldsymbol{e})_i, \boldsymbol{y}_i) \leq \boldsymbol{\epsilon}_{y_i}\} \geq p. \tag{25}$$

*Considering bounded output cases, i.e., $\boldsymbol{l} \leq \boldsymbol{f}_\theta(\boldsymbol{z}) \leq \boldsymbol{u}, \forall \boldsymbol{z}$ subject to $\boldsymbol{l} \leq \boldsymbol{l}_b \leq \boldsymbol{u}_b \leq \boldsymbol{u}$, and if for those samples which are accepted by the user, we have $|\mathbb{E}\{\boldsymbol{f}_\theta(\boldsymbol{x} + \boldsymbol{\delta} + \boldsymbol{e})\}_i - \boldsymbol{f}_\theta(\boldsymbol{x})_i| \leq \tau$, $0 \leq \tau \leq \min(\boldsymbol{f}_\theta(\boldsymbol{x}) - \boldsymbol{l}_b, \boldsymbol{u}_b - \boldsymbol{f}_\theta(\boldsymbol{x}))$, for the convex accepted region (set) and as $n \to \infty$, the following inequality holds $\forall i \in [t]$*

$$\mathbb{P}\{diss_y(\boldsymbol{g}_n(\boldsymbol{x} + \boldsymbol{\delta})_i, \boldsymbol{y}_i) \leq \boldsymbol{\epsilon}_{y_i}\} \geq$$
$$\min_{i \in [t]} \begin{cases} I_p(\lceil n(1 - \frac{\boldsymbol{u}_{b_i} - \boldsymbol{f}_\theta(\boldsymbol{x})_i - \tau}{\boldsymbol{u}_i - \boldsymbol{f}_\theta(\boldsymbol{x})_i - \tau})\rceil, \lceil n\frac{\boldsymbol{u}_{b_i} - \boldsymbol{f}_\theta(\boldsymbol{x})_i - \tau}{\boldsymbol{u}_i - \boldsymbol{f}_\theta(\boldsymbol{x})_i - \tau}\rceil + 1), & if \ \frac{\boldsymbol{u}_i - \boldsymbol{u}_{b_i}}{\boldsymbol{u}_i - \boldsymbol{f}_\theta(\boldsymbol{x})_i - \tau} \geq \frac{\boldsymbol{l}_i - \boldsymbol{l}_{b_i}}{\boldsymbol{f}_\theta(\boldsymbol{x})_i - \tau - \boldsymbol{l}_i} \\ I_p(\lceil n(1 - \frac{\boldsymbol{f}_\theta(\boldsymbol{x})_i - \tau - \boldsymbol{l}_{b_i}}{\boldsymbol{f}_\theta(\boldsymbol{x})_i - \tau - \boldsymbol{l}_i})\rceil, \lceil n(\frac{\boldsymbol{f}_\theta(\boldsymbol{x})_i - \tau - \boldsymbol{l}_{b_i}}{\boldsymbol{f}_\theta(\boldsymbol{x})_i - \tau - \boldsymbol{l}_i})\rceil + 1), & otherwise, \end{cases} \tag{26}$$

where $I_p(a, b)$ is the regularized incomplete beta function defined as $I_p(a, b) = \frac{1}{B(a,b)} \int_0^p t^{a-1}(1 - t)^{b-1} dt$ and $B(a, b)$ is the complete beta function.

*Proof.* Considering the worst-case scenario to find the lower bound on the probability of estimating a valid $\mathbf{g}_n$ suggests considering the output generated by $\mathbf{f}_\theta(\mathbf{x} + \boldsymbol{\delta} + \mathbf{e})_i$ follows a Bernoulli distribution where the outcome is located with probability $p$ at $\mathbf{f}_\theta(\mathbf{x})_i + \tau$ (successful event and true as $n \to \infty$) and with probability $1 - p$ at $\mathbf{u}_i$ (unsuccessful event) if $\frac{\mathbf{u}_i - \mathbf{u}_{b_i}}{\mathbf{u}_i - \mathbf{f}_\theta(\mathbf{x})_i - \tau} \geq \frac{\mathbf{l}_i - \mathbf{l}_{b_i}}{\mathbf{f}_\theta(\mathbf{x})_i - \tau - \mathbf{l}_i}$ (see Figure 9), otherwise, at $\mathbf{f}_\theta(\mathbf{x})_i - \tau$ and $\mathbf{l}_i$. Without loss of generality let us assume for $i^{th}$ output, the former condition is satisfied. Now by defining the random variable $O$ as the number of valid outputs, we have

$\mathbb{P}\{\text{diss}_y(\mathbf{g}_n(\mathbf{x} + \boldsymbol{\delta})_i, \mathbf{y}_i) \leq \epsilon_{y_i}\}$

$$
\begin{aligned}
&= \sum_{v=0}^{n} \mathbb{P}\{\text{diss}_y(\mathbf{g}_n(\mathbf{x} + \boldsymbol{\delta})_i, \mathbf{y}_i) \leq \epsilon_{y_i} | O = v\} \mathbb{P}\{O = v\} \\
&\geq \sum_{v=n(1 - \frac{\mathbf{u}_{b_i} - \mathbf{f}_\theta(\mathbf{x})_i - \tau}{\mathbf{u}_i - \mathbf{f}_\theta(\mathbf{x})_i - \tau})}^{n} \binom{n}{v} (p)^v (1 - p)^{n-v} \\
&\geq 1 - \sum_{v=0}^{\lceil n(1 - \frac{\mathbf{u}_{b_i} - \mathbf{f}_\theta(\mathbf{x})_i - \tau}{\mathbf{u}_i - \mathbf{f}_\theta(\mathbf{x})_i - \tau})\rceil} \binom{n}{v} (p)^v (1 - p)^{n-v} \\
&= \sum_{v=0}^{\lceil n\frac{\mathbf{u}_{b_i} - \mathbf{f}_\theta(\mathbf{x})_i - \tau}{\mathbf{u}_i - \mathbf{f}_\theta(\mathbf{x})_i - \tau}\rceil} \binom{n}{v} (1 - p)^v (p)^{n-v} \\
&= I_p(\lceil n(1 - \frac{\mathbf{u}_{b_i} - \mathbf{f}_\theta(\mathbf{x})_i - \tau}{\mathbf{u}_i - \mathbf{f}_\theta(\mathbf{x})_i - \tau})\rceil, \lceil n\frac{\mathbf{u}_{b_i} - \mathbf{f}_\theta(\mathbf{x})_i - \tau}{\mathbf{u}_i - \mathbf{f}_\theta(\mathbf{x})_i - \tau}\rceil + 1).
\end{aligned}
$$

This is a lower bound for the probability of one of the outputs being in the accepted region if $\frac{\mathbf{u}_i - \mathbf{u}_{b_i}}{\mathbf{u}_i - \mathbf{f}_\theta(\mathbf{x})_i - \tau} \geq \frac{\mathbf{l}_i - \mathbf{l}_{b_i}}{\mathbf{f}_\theta(\mathbf{x})_i - \tau - \mathbf{l}_i}$. For the other condition, we only replace $\mathbf{u}_{b_i}$ and $\mathbf{u}_i$ with $\mathbf{l}_{b_i}$ and $\mathbf{l}_i$ and take care of signs. To derive the same lower bound for the entire output variables, one can take a minimum over all these probability values to complete the proof. □

## E Proof of Non-Asymptotic Certification of Averaging Function for Bounded Outputs

We repeat the theorem's statement here for convenience, followed by its proof.

**Proposition 1:** (Discounted Certification of $\mathbf{g}(\mathbf{x})$ against $\ell_2$ Attack for Bounded Outputs). *Let $\boldsymbol{f}_\theta(\boldsymbol{x})$ : $\mathbb{R}^d \to \mathbb{R}^t$ be a (possibly) randomized base regressor and suppose outputs generated by $\boldsymbol{f}_\theta(\boldsymbol{x})$ are independent and identically distributed with a user-defined $\boldsymbol{\epsilon}_y$ (equivalent to $\boldsymbol{u}_b$ and $\boldsymbol{l}_b$) to define the accepted region. Suppose for $\boldsymbol{e} \sim \mathcal{N}(\boldsymbol{0}, \sigma^2 \boldsymbol{I})$, $\forall \|\boldsymbol{\delta}\|_2 \leq \epsilon_x$, and an arbitrary value of $p$, as stated in (4), $\forall i \in [t]$:*

$$\mathbb{P}\{diss_y(\boldsymbol{f}_\theta(\boldsymbol{x} + \boldsymbol{\delta} + \boldsymbol{e})_i, \boldsymbol{y}_i) \leq \boldsymbol{\epsilon}_{y_i}\} \geq p. \tag{27}$$

*Considering bounded output cases, i.e., $\boldsymbol{l} \leq \boldsymbol{f}_\theta(\boldsymbol{z}) \leq \boldsymbol{u}, \forall \boldsymbol{z}$ subject to $\boldsymbol{l} \leq \boldsymbol{l}_b \leq \boldsymbol{u}_b \leq \boldsymbol{u}$ and assuming the accepted region (set) for each output target variable to be convex, then given a discount factor $\beta \geq 0$ such that $\boldsymbol{l} \leq \boldsymbol{l}_b - \beta|\boldsymbol{f}_\theta(\boldsymbol{x}) - \boldsymbol{l}_b| \leq \boldsymbol{u}_b + \beta|\boldsymbol{f}_\theta(\boldsymbol{x}) - \boldsymbol{u}_b| \leq \boldsymbol{u}$ holds, then the following inequality holds for $\forall i \in [t]$*

$$
\begin{aligned}
&\mathbb{P}\{diss_y(\boldsymbol{g}_n(\boldsymbol{x} + \boldsymbol{\delta})_i, \boldsymbol{y}_i) \leq (1 + \beta)\boldsymbol{\epsilon}_{y_i}\} \geq \\
&\min_{i \in [t]} \begin{cases} I_p(\lceil n(1 - \frac{\beta|\boldsymbol{u}_{b_i} - \boldsymbol{f}_\theta(\boldsymbol{x})_i|}{\boldsymbol{u}_i - \boldsymbol{u}_{b_i}})\rceil, \lceil n(\frac{\beta|\boldsymbol{u}_{b_i} - \boldsymbol{f}_\theta(\boldsymbol{x})_i|}{\boldsymbol{u}_i - \boldsymbol{u}_{b_i}})\rceil + 1), & if \ \frac{|\boldsymbol{u}_{b_i} - \boldsymbol{f}_\theta(\boldsymbol{x})_i|}{\boldsymbol{u}_i - \boldsymbol{u}_{b_i}} \leq \frac{|\boldsymbol{l}_{b_i} - \boldsymbol{f}_\theta(\boldsymbol{x})_i|}{\boldsymbol{l}_{b_i} - \boldsymbol{l}_i} \\ I_p(\lceil n(1 - \frac{\beta|\boldsymbol{l}_{b_i} - \boldsymbol{f}_\theta(\boldsymbol{x})_i|}{\boldsymbol{l}_{b_i} - \boldsymbol{l}_i})\rceil, \lceil n\frac{\beta|\boldsymbol{l}_{b_i} - \boldsymbol{f}_\theta(\boldsymbol{x})_i|}{\boldsymbol{l}_{b_i} - \boldsymbol{l}_i}\rceil + 1), & otherwise, \end{cases}
\end{aligned} \tag{28}
$$

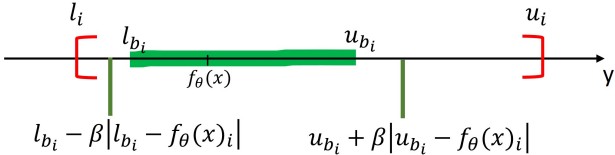

Figure 10: The upper and lower bounds of $i^{th}$ output variable are denoted by $\mathbf{u}_i$ and $\mathbf{l}_i$, and the upper and lower bounds of the acceptable region for the base regression model are shown by green colour between $\mathbf{u}_{b_i}$ and $\mathbf{l}_{b_i}$. In the discounted certificate case, the user is asked to extend this acceptable region utilizing new bounds and the discount factor $\beta$ as shown in the figure.

*where $I_p(a, b)$ is the regularized incomplete beta function defined as $I_p(a, b) = \frac{1}{B(a,b)} \int_0^p t^{a-1}(1-t)^{b-1} dt$ and $B(a, b)$ is the complete beta function.*

*Proof.* The proof is similar to the proof of Theorem 3 where the lower and upper bounds of the accepted region are replaced with the new discounted bounds. Considering the worst-case scenario to find the lower bound on the probability of estimating a valid $\mathbf{g}_n$ suggests considering the output generated by $\mathbf{f}_\theta(\mathbf{x} + \boldsymbol{\delta} + \mathbf{e})_i$ follows a Bernoulli distribution where the outcome is located with probability $p$ at $\mathbf{u}_{b_i}$ (successful event) and with probability $1 - p$ at $\mathbf{u}_i$ (unsuccessful event) if $\frac{|\mathbf{u}_{b_i} - \mathbf{f}_\theta(\mathbf{x})_i|}{\mathbf{u}_i - \mathbf{u}_{b_i}} \leq \frac{|\mathbf{l}_{b_i} - \mathbf{f}_\theta(\mathbf{x})_i|}{\mathbf{l}_{b_i} - \mathbf{l}_i}$ (see Figure 10), otherwise, at $\mathbf{l}_{b_i}$ and $\mathbf{l}_i$. Without loss of generality let us assume for $i^{th}$ output, the former condition is satisfied. Now by defining the random variable $O$ as the number of valid outputs, we have

$$\mathbb{P}\{\text{diss}_y(\mathbf{g}_n(\mathbf{x} + \boldsymbol{\delta})_i, \mathbf{y}_i) \leq (1 + \beta)\epsilon_{y_i}\}$$

$$= \sum_{v=0}^{n} \mathbb{P}\{\text{diss}_y(\mathbf{g}_n(\mathbf{x} + \boldsymbol{\delta})_i, \mathbf{y}_i) \leq (1 + \beta)\epsilon_{y_i} | O = v\} \mathbb{P}\{O = v\}$$

$$\geq \sum_{v=n(1 - \frac{\beta|\mathbf{u}_{b_i} - \mathbf{f}_\theta(\mathbf{x})_i|}{\mathbf{u}_i - \mathbf{u}_{b_i}})}^{n} \binom{n}{v} (p)^v (1-p)^{n-v}$$

$$\geq 1 - \sum_{v=0}^{\lceil n(1 - \frac{\beta|\mathbf{u}_{b_i} - \mathbf{f}_\theta(\mathbf{x})_i|}{\mathbf{u}_i - \mathbf{u}_{b_i}}) \rceil} \binom{n}{v} (p)^v (1-p)^{n-v}$$

$$= \sum_{v=0}^{\lceil n \frac{\beta|\mathbf{u}_{b_i} - \mathbf{f}_\theta(\mathbf{x})_i|}{\mathbf{u}_i - \mathbf{u}_{b_i}} \rceil} \binom{n}{v} (1-p)^v (p)^{n-v}$$

$$= I_p(\lceil n(1 - \frac{\beta|\mathbf{u}_{b_i} - \mathbf{f}_\theta(\mathbf{x})_i|}{\mathbf{u}_i - \mathbf{u}_{b_i}}) \rceil, \lceil n(\frac{\beta|\mathbf{u}_{b_i} - \mathbf{f}_\theta(\mathbf{x})_i|}{\mathbf{u}_i - \mathbf{u}_{b_i}}) \rceil + 1).$$

This is a lower bound for the probability of one of the outputs being in the accepted region if $\frac{|\mathbf{u}_{b_i} - \mathbf{f}_\theta(\mathbf{x})_i|}{\mathbf{u}_i - \mathbf{u}_{b_i}} \leq \frac{|\mathbf{l}_{b_i} - \mathbf{f}_\theta(\mathbf{x})_i|}{\mathbf{l}_{b_i} - \mathbf{l}_i}$. For the other condition, we only replace $\mathbf{u}_{b_i}$ and $\mathbf{u}_i$ with $\mathbf{l}_{b_i}$ and $\mathbf{l}_i$ and take care of signs. To derive the same lower bound for the entire output variables, one can take a minimum over all these probability values to complete the proof. $\square$

## F Estimated Radius vs P

Figure 11 illustrates the input upper bound on the maximum certifiable perturbation as a function of the user-defined probability. If the user asks for a larger portion of valid outputs, the $\ell_2$-ball around the base prediction gets smaller.

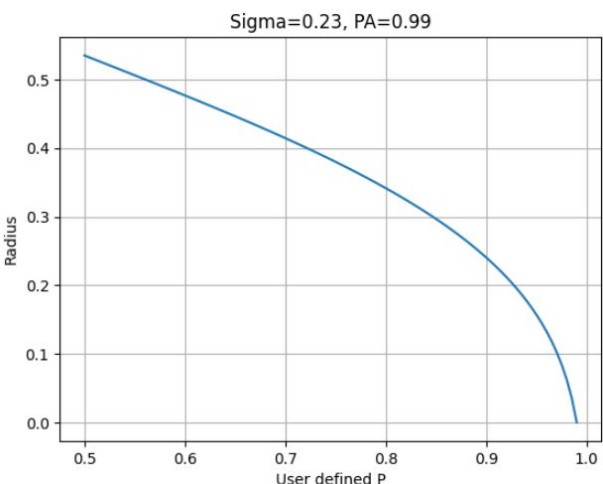

Figure 11: Adversarial upper bound vs user defined probability.

