# OpenReview forum: "RS-Reg: Probabilistic and Robust Certified Regression through Randomized Smoothing"
_TMLR — Accepted by TMLR_

### Review · Reviewer_ESMF · 2025-02-01

**Summary Of Contributions:**

This paper proposes an extension of the randomized smoothing (RS) approach to certifiable adversarial robustness to regression tasks. The authors propose a definition of probabilistic robustness in this new setting, show the type of guarantee a standard perturbation-based approach yields (Theorem 1), and then discusses a new approach in which an average over the outputs of randomly perturbed inputs is taken (Theorems 2 and 3, and Proposition 1). The paper concludes with experimental results demonstrating the certification on a synthetic regression task and on a computer vision task.

**Audience:**

Yes

**Broader Impact Concerns:**

Broader impacts are sufficiently addressed in the submission.

**Claims And Evidence:**

No

**Requested Changes:**

1. A discussion of directions for extending the guarantees beyond l2 perturbations. Are there naive ways to do so? Is there a principled geometric approach?
2. In the third bullet point of Section 1, it’s not very clear without reading the paper what “RS leads to [a] multivariate normal density” means, although point (ii) in the paragraph immediately below is clearer about this.
3. In the last paragraph of Section 1, it is not clear that the asymptotic variable is the number of samples from a Gaussian used to perturb the input and then average the output.
4. “recent study Hammoudeh” -> “recent study by Hammoudeh”
5. Are there notable applications that you can cite for certified regression outside the case of object detection where the target is the Softmax?
6. Fixing some uses of \citet instead of \citep (e.g. in the Randomized Smoothing part of Section 3).
7. Definition 1: “If” -> “if”
8. Theorem 1: “definition Eq. (2)” -> “Definition 1”
9. The discussion around Theorem 1 is confusing to me: in the paragraph before it says it is about an arbitrary model that has _not_ undergone randomized smoothing, but in the theorem itself the input to the model is perturbed by a vector e in the exact same way that randomized smoothing operates. How is the theorem _not_ about a function that has undergone randomized smoothing?
10. [critical] The meaning of Theorem 2 is very hard to parse. It seems to take n to infinity, but the primary equation (5) has n in it. Furthermore, in the remarks below it would be helpful to have an explainer for how to interpret the double summation in equation 5, and specifically the effect of the subsets in R_k.
11. [critical] Similarly, Theorem 3 and Proposition 1 are challenging to parse and involve unclear asymptotic notation.

**Strengths And Weaknesses:**

## Strengths
1. The paper formally studies the extension of RS to regression tasks, a critical and understudied (in this context) area of machine learning.
2. The approach of using averaging instead of just randomization for certifying robustness in this way seems novel and may be useful in settings where a deterministic classifier is desirable.
3. The authors verify their results via a thoroughly analyzed set of experiments in both a synthetic data setting and a more practical task from computer vision. They also promise to provide code to reproduce these results.

## Weaknesses
1. While Theorem 1 is fairly clear, Theorems 2 and 3 are not stated in a way that allows for clear insights into the dependence of the bounds on the underlying problem; it seems the focus is only on enabling the bounds’ computation (which is useful, but less clear). While the remarks following these theorems help somewhat, it is hard to understand how the probability depends on the allowed perturbation in a clear, functional way. Corollaries about special cases following the theorems would be useful to the reader.
2. Theorems 2 and 3 (as well as Proposition 1) make use of a confusing asymptotic setting where the limit as n approaches infinity of a quantity that depends on n is yet another quantity that depends on n. It is reasonable to say that the limit of an (n-dependent) probability is the limit of another (n-dependent) probability, but it is not clear what it means when the limit of an (n-dependent) probability is an n-dependent function that becomes 1 when n goes to infinity. From the proof I believe the issue is direct substitution of the n-dependent CLT expression rather than working with limits.
3. The non-asymptotic averaging seems crucial for any practical implementation of this method, but it is not thoroughly discussed and requires additional assumptions on top of the (already possibly unrealistic, given the application to softmax object detection) boundedness assumptions.
4. There are not very many motivations provided in terms of applications.
5. There is a definition provided for certification in the regression case, but the theorems do not make use of this definition (even if they do imply that the models satisfy it).
6. It is not clear to me that norm-based perturbation is the right model for practical adversarial robustness, although such results could end up being useful beyond the scope of the paper.

## Summary
I think the paper is of sufficient interest to the ML community to be accepted at TMLR after addressing the issues with the presentation of Theorems 2 and 3.

---

> ### Author Response · Authors · 2025-02-18
> **Response to Reviewer ESMF, Part 1**
>
> The authors would like to thank the reviewer for dedicating their time to reading our manuscript. We acknowledge the reviewer's observation that the concept behind randomized smoothing in regression is a critical but understudied. Below, we provide a detailed response to all questions and suggestions made by the reviewer.
>
> **1. A discussion of directions for extending the guarantees beyond l2 perturbations. Are there naive ways to do so? Is there a principled geometric approach?**
>
> Thank you for this important question. Indeed, in the Introduction, we briefly stated: *``While these guarantees are constrained to a certain type of threat model (which may not align with the attacker’s actual strategy), they offer precise and reliable bounds crucial for cybersecurity systems.’’* While we acknowledge limitations of the bounded-$\ell_2$ threat model, this has become canonical in the attack and certified defense literature. There are historical reasons for this: it matches the Gaussian smoothing mechanism, with density that is exponential in the $\ell_2$. Given it is by far and away the most popular threat model for certifying classifiers, we believe this to be an important focus (at least initially) in regression.
>
> Concretely, the $\ell_2$ is a reasonable surrogate for related threat models such as the $\ell_1$ (e.g., total perturbation to pixels in an input image) or $\ell_\infty$ (e.g., each input attribute its own independent perturbation budget). These approximate natural requirements on attackers that their perturbations be stealthy to human oversight of model predictions, or stand in for a domain-specific cost model of the attacker who might need to retain some payload in an input while circumventing a defense.
>
> A possible application is to certification in the feature space rather than input space. It’s quite reasonable to assume perturbations in the feature space can be in additive form and suitable to be modelled by existing norms. We are also aware of some verification techniques that consider convolutional perturbations such as blurring or sharpening for image data [a].
>
> We believe the community is actively working to extend guarantees to a wider range of threat models. A straightforward extension of the $\ell_2$ norm is to consider other widely used norms in the input space, such as $\ell_1$, $\ell_p$, and $\ell_\infty$ norms. Beyond the aforementioned motivations for these threat models, popular exponential family density functions are related to these norms.
>
> [a] Brückner, Benedikt, and Alessio Lomuscio. "Verification of Neural Networks against Convolutional Perturbations via Parameterised Kernels." arXiv preprint arXiv:2411.04594 (2024).
>
> We have added to the revised manuscript: *Although this paper focuses on the $\ell_2$ threat model, one can extend the results to other norms such as $\ell_1$, $\ell_p$, and $\ell_\infty$, either in the input or feature space. Additionally, one could explore scenarios where perturbations are applied to parameter space in convolutional kernels, such as in blurring or sharpening operators in the context of image modality.*
>
> **2. In the third bullet point of Section 1, it’s not very clear without reading the paper what “RS leads to [a] multivariate normal density” means, although point (ii) in the paragraph immediately below is clearer about this.**
>
> We have rephrased the third bullet point to convey the message in the revised manuscript: *``We then demonstrate that, asymptotically, the output of RS with averaging follows a normal distribution, allowing the probability of obtaining valid results from a smoothed regressor to be determined through an integral over a neighbourhood of such a normal density.’’* We hope that this clarifies the meaning.
>
> **3. In the last paragraph of Section 1, it is not clear that the asymptotic variable is the number of samples from a Gaussian used to perturb the input and then average the output.**
>
> Thanks for pointing this out, the following sentence has been added to the revised manuscript: *``… as $n \rightarrow \infty$ (where $n$ is the number of samples drawn from a Gaussian used to perturb the input and then averaged to compute the output) …’’*
>
> **4. “recent study Hammoudeh” -> “recent study by Hammoudeh”**
>
> This typo has been fixed in the revised manuscript.
>
> **5. Are there notable applications that you can cite for certified regression outside the case of object detection where the target is the Softmax?**
>
> To the best of our knowledge, apart from the method proposed by Chiang et al. (2020) in the context of object detection, no other certification technique has demonstrated the use of a Softmax-based regression network for other applications. Since a certification model for regression tasks should be model agnostic, incorporating such a layer could restrict the applicability of the certification approach, and this was the main motivation of the present study.

---

> ### Author Response · Authors · 2025-02-18
> **Response to Reviewer ESMF, Part 2**
>
> **6. Fixing some uses of \citet instead of \citep (e.g. in the Randomized Smoothing part of Section 3).**
>
> **7. Definition 1: “If” -> “if”**
>
> **8. Theorem 1: “definition Eq. (2)” -> “Definition 1”**
>
> These typos have been fixed in the revised manuscript.
>
> **9. The discussion around Theorem 1 is confusing to me: in the paragraph before it says it is about an arbitrary model that has not undergone randomized smoothing, but in the theorem itself the input to the model is perturbed by a vector e in the exact same way that randomized smoothing operates. How is the theorem not about a function that has undergone randomized smoothing?**
>
> We hope to clarify this below. In the context of regression, randomized smoothing, as defined in equation (6), averages the outputs of $n$ perturbed versions of the input $\textbf{x}$, which is why it is referred to as ``smooth’’—it effectively smooths out variations across different observations. Now, consider the case where $n=1$, meaning no smoothing is applied, and we analyze the output behaviour based on a single perturbed sample of $\textbf{x}$. In this scenario, all discussions hold for the base function$\textbf{f}_\theta(.)$. The input remains randomized, which is the only reason why the guarantee is probabilistic, even if the underlying regression model itself is deterministic.
>
> **10. [critical] The meaning of Theorem 2 is very hard to parse. It seems to take n to infinity, but the primary equation (5) has n in it. Furthermore, in the remarks below it would be helpful to have an explainer for how to interpret the double summation in equation 5, and specifically the effect of the subsets in R_k.**
>
> Thanks for raising this point. We are not claiming that $n$ disappears or that we are taking a strict limit as $n \rightarrow \infty$. Instead, we are describing the asymptotic behavior for large $n$, where the expression simplifies while still being n-dependent. This is similar to the familiar Central Limit Theorem, where distributions approximate a normal form for large $n$ but still retain dependence on $n$. To aid in understanding this result, we simplified the quantity of Theorem 2 for the case where $t=2$. We now present this as an example immediately following Theorem 2.
> Example 1: For clarity and better interoperability of the results in Theorem 2, let's consider the case where $t=2$, representing a regression model with two output variables. Then the quantity in (5) reduces to $\boldsymbol{\Phi}({\sqrt{n}}\hat{\boldsymbol{\Sigma}}^{-\frac{1}{2}}(u_b))-g_n(x+\boldsymbol{\delta})))-\boldsymbol{\Phi}({\sqrt{n}}\hat{\boldsymbol{\Sigma}}^{-\frac{1}{2}}([l_{b_1}, u_{b_2}]^\top-g_n(x+\boldsymbol{\delta})))-\boldsymbol{\Phi}({\sqrt{n}}\hat{\boldsymbol{\Sigma}}^{-\frac{1}{2}}([u_{b_1}, l_{b_2}]^\top-g_n(x+\boldsymbol{\delta})))+\boldsymbol{\Phi}({\sqrt{n}}\hat{\boldsymbol{\Sigma}}^{-\frac{1}{2}}(l_b-g_n(x+\boldsymbol{\delta})))$. As can be observed, the first summation in (5) only takes care of signs and the second summation constructs the terms based on lower/upper bounds formed by $\textbf{c}_\mathcal{D}$, to accurately compute the probability without omissions or redundant calculations of any region.
>
> **Disclosure**: We did not use the results from Theorem 2 in any subsequent theorems or elsewhere in the paper. It was included solely to provide intuition for the reader, illustrating that the Central Limit Theorem does not significantly aid in this context as explained in follow-up Remarks—something that was not immediately obvious to us at first—despite our use of averaging in a large-number regime.
>
> **11. [critical] Similarly, Theorem 3 and Proposition 1 are challenging to parse and involve unclear asymptotic notation.**
>
> The use of $n$ is similar to its use in Theorem 2, where we do not use it as a strict limit as $n \rightarrow \infty$. Rather, we are describing the behavior for large $n$, where the expression simplifies while still remaining n-dependent. Although the expression in Theorem 3 may seem unintuitive without understanding the broader context, we believe it becomes clear when reading its proof in Appendix D, as explained in Figure 9 and through the geometric interpretation of the problem. Unfortunately, restating the result as a Corollary for a univariate model does not provide additional insight in this case, as it merely removes the min operation while keeping the remaining terms unchanged. We’ve found the approach that works for us is to carefully follow the proof and understand the role of each parameter involved. If this doesn’t assist, we’d welcome other suggestions. We’re also happy to reorder the proof into the main body if that is deemed worthwhile.
>
> Unlike other results, Proposition 1 addresses the non-asymptotic case, so it is expected that $n$ remains explicitly present. However, we acknowledge that these types of results are complex. In practice, once applied in simulations or experiments, they are straightforward to compute.

---

### Review · Reviewer_ovzF · 2025-02-03

**Summary Of Contributions:**

This paper introduces a randomized smoothing-based framework for regression models. In particular, it proposes a definition of a probabilistic robustness certificate for regression tasks (Definition 1) and a series of theoretical analysis results for deriving the robustness certificate under asymptotic and non-asymptotic (bounded) output cases. Empirical evaluations on a simulation dataset and a camera re-location task are provided.

**Audience:**

Yes

**Broader Impact Concerns:**

No broader impact concerns

**Claims And Evidence:**

No

**Requested Changes:**

1. Explain how your work contributes to the field with reference to the NeurIPS work of [1].

2. Highlight more on the technical challenges you encountered in proving the theoretical results in Section 3. Given the seminal randomized smoothing work of Cohen et al. (2019), the technical contributions of this work are not clear, especially given the several assumptions imposed in the technical theorems.

3. Did you compare your robustness certificate with empirical attacks developed by Chiang et al. (2020) and Liu et al. (2022)? How tight is your robustness certificate?

4. Improve the presentation of Section 3 by providing clearer explanations of the mathematical notations and detailed discussions of the assumptions.

5. Conduct more comprehensive experimental results to benchmark the proposed method's performance and compare it to related methods.

**Strengths And Weaknesses:**

The strengths of this paper lie in the rigorous theoretical analyses of how randomized smoothing can be applied to regression tasks. However, a major issue of this paper is its large text overlap with a NeurIPS 2024 paper [1] (not cited anywhere in the manuscript), including problem setup, certification framework, and experimental results. In my opinion, [1] provides a more general certification method for regression models, covering both bounded and unbounded output settings. As a result, I do not see how this paper contributes to the field with reference to [1], despite the fact that [1] cited this paper as a preliminary baseline.

The experimental results of this paper are fairly limited. In the related work section, you mentioned a few existing works (Pautov et al., 2022a; Salman et al., 2019; Chiang et al., 2020) that extend randomized smoothing outside classification, but they impose different constraints on the regression outputs or use different techniques. It is not clear why the authors do not conduct comparison studies with these related methods. Figure 2 and Figure 3 are of low resolution, which also takes up much space than necessary. I feel the experimental figures presented are difficult to parse, and there is a lack of clear explanations about the experimental settings (for general regression tasks and for the specific tasks considered in the experiments). In addition, I found it difficult to understand the introduced terminologies from place to place, especially the mathematical notations and the technical assumptions introduced in Section 4.

[1] Certified Adversarial Robustness via Randomized α-Smoothing for Regression Models, Rekavandi et al., NeurIPS 2024, https://openreview.net/pdf/2c1928be34a76219c66d9466b72ff8cde2c15a4e.pdf

---

> ### Author Response · Authors · 2025-02-18
> **Response to Reviewer ovzF**
>
> The authors thank the reviewer for taking the time to read our manuscript. We recognize the reviewer's perspective on the significance of rigorous theoretical analyses in applying randomized smoothing to regression tasks.
>
> **1. Explain how your work contributes to the field with reference to the NeurIPS work of [1].**
>
> We have evidence that establishes [1] as following this present work. We have shared this with the editor in confidence, who may decide when/how to reveal this to reviewers.
>
> [1] Certified Adversarial Robustness via Randomized α-Smoothing for Regression Models, Rekavandi et al., NeurIPS 2024.
>
> **2. Highlight more on the technical challenges you encountered in proving the theoretical results in Section 3. Given the seminal randomized smoothing work of Cohen et al. (2019), the technical contributions of this work are not clear, especially given the several assumptions imposed in the technical theorems.**
>
> Resemblance of Theorem 1 with the seminal work of Cohen et al. (2019) was intentional to facilitate interpretation and comparison with existing results. The key distinction of our work lies in its focus on regression models, where the output is continuous, as opposed to classification tasks with categorical outputs.
>
> In the classification setting of Cohen et al., the smoothing function is based on majority voting, meaning each output observation contributes equally, and the category with the highest vote count determines the final decision. By contrast, regression outputs are continuous, and their influence in averaging is weighted by their magnitude. As a result, a single observation can disproportionately affect the outcome if it is significantly larger than others, and this was the main challenge of studying smoothed regressed values, which involved worst-case scenario analysis.
>
> To address this, Theorem 2 examines the asymptotic behavior of the averaged value, while Theorem 3 and Proposition 1 analyze the averaging process under specific model assumptions or constraints on the acceptance range, depending on the application in which certification is being performed. These results are fundamentally different from those of Cohen et al. (2019), both in theory and practical application, despite some similarities in notation.
>
> **3. Did you compare your robustness certificate with empirical attacks developed by Chiang et al. (2020) and Liu et al. (2022)? How tight is your robustness certificate?**
>
> The works of Chiang et al. (2020) and Liu et al. (2022) focus on attack and defense strategies in the context of object detection, which is an application that is outside the scope of our study and their analysis was not applicable to our model under the investigation because of their assumptions. Their choice to report results on object detection may stem from the nature of the certificates they analyze, which are derived from scaled versions of the Softmax layer. By contrast, our analysis imposes no constraints on any specific layers of the regression model—it applies to any model architecture.
>
> Instead, we explore the application of our framework in the context of visual positioning systems, where certification is crucial for enhancing safety in autonomous driving. Figure 4 empirically demonstrates the tightness of our certificate. For the base model, the observed success rate is only slightly above the expected level, indicating that the certificate is tight. However, for the averaged quantity, the certificates are intentionally conservative, as they are designed for worst-case scenarios to ensure they are never violated. This was discussed in the result section.
>
> Future research could focus on improving the tightness of these certificates, potentially by exploring alternative smoothing techniques.
>
> **4. Improve the presentation of Section 3 by providing clearer explanations of the mathematical notations and detailed discussions of the assumptions.**
>
> We have provided notation used in the paper in the **Notation** paragraph of Section 3. We would gladly refine notation and assumptions further if you could please let us know which ones are unclear or ambiguous.
>
> **5. Conduct more comprehensive experimental results to benchmark the proposed method's performance and compare it to related methods.**
>
> We appreciate the suggestion for further experiments. We feel that the paper offers a range of contributions: algorithms (new mechanisms), theoretical development of sound certificates and theoretical analysis of these, as well as experimental demonstration. As mentioned, the other certificates are not practical for our case study as our model does not have softmax layer in the end and therefore their results are not correct. Instead, we compared our own results in different parameter settings with the base model. As such we did not elect to conduct these particular experiments for this work.

---

### Review · Reviewer_avtB · 2025-02-09

**Summary Of Contributions:**

This paper is an extension of randomized smoothing to regression tasks. The general idea of random smoothing is that by convolving a predictor with some noise, it can be made smoother. If sufficiently smooth, it will be difficult for an adversary to flip the outputs by perturbing the function.

In the most traditional model, the task is classification, the noise is Gaussian, the adversary is assumed to be bounded to perturb inputs within an l2 ball, and the resulting smoothness is Lipschitz in an l2 sense. There are various elaborations on this framework considering different geometries for the adversary and therefore different types of noise, as well as attacks such as patch attacks that can’t be characterized as lying within some metric ball. The most related work, which the authors point out, is by Chiang et al., which reduces bound-box object detection to regression, and then applies a median-based smoothing approach. The results here are quite different.

The authors here focus on regression tasks (possibly multi-dimensional), and they consider a slightly different threat model — the adversary still perturbs within an l2 ball, but now the requirement for robustness of the outputs is if they do move outside some distance, this happens with a low-enough probability.

The authors give a randomized smoothing approach that works under this threat model. Some technical differences arise from the classification case, because the exact location of the true output may vary (unlike in the classification case where the goal of the adversary is just to flip a class). The authors consider the very reasonable special case of bounded outputs (where the user wants even tighter bounds than are known a priori) to deal with this problem. In experiments, they show that one can adversarially-train a classifier and then apply smoothing to get practically effective results.

**Audience:**

Yes

**Broader Impact Concerns:**

No broader impact concerns.

**Claims And Evidence:**

Yes

**Requested Changes:**

I think the explanation of the discounting result (Proposition 1) would be helpful, as I had a hard time understanding this part of the paper and what the discount is supposed to mean about the user's threat model. It also couldn't hurt to actually write out the proof (even though almost the same as another proof) in the appendices.

**Strengths And Weaknesses:**

The strength is that the results are solid and well-supported, so that I don't have substantive complaints.

The main weakness is that the paper is a little terse and hard to follow sometimes.

---

> ### Author Response · Authors · 2025-02-18
> **Response to Reviewer avtB**
>
> The authors thank the reviewer for carefully reading and evaluating our manuscript. We are grateful for the recognition of our results as both sound and novel in the context of the literature.
>
> **1. I think the explanation of the discounting result (Proposition 1) would be helpful, as I had a hard time understanding this part of the paper and what the discount is supposed to mean about the user's threat model.**
>
> We have reworded and integrated the explanation from the appendix about this proposition with our discussion in the main text more clearly conveying the concept and addressing the scenario analyzed. To clarify the results further, Figure 10 in the appendix illustrates the geometry of the problem for one of the output variables.
>
> **2. It also couldn't hurt to actually write out the proof (even though almost the same as another proof) in the appendices.**
>
> As suggested, this proof has been added to the revised manuscript (Appendix E).

---

### Decision · Action_Editor_i78u · 2025-04-09

**Recommendation:** Accept as is

**Comment:**

The relevant details are discussed above.

**Audience:**

There was some concern about the relationship to a paper that was posted to arxiv later than this submission but was published earlier than this submission.  After consultation with an EIC, the decision was to evaluate the submission in context of that existing published work.  The reviewers agreed that there is enough new material relative to the published work that this paper woudl still be of interest to members of TMLR's audience.

**Claims And Evidence:**

The claims are convincingly supported by evidence.